# The bacterial replication origin BUS promotes nucleobase capture

Simone Pelliciari[1], Salomé Bodet-Lefèvre [2], Stepan Fenyk[1], Daniel Stevens [1], Charles Winterhalter [1], Frederic D. Schramm [1], Sara Pintar [1], Daniel R. Burnham[3], George Merces[1], Tomas T. Richardson [1], Yumiko Tashiro[2], Julia Hubbard[1], Hasan Yardimci [3], Aravindan Ilangovan [2] ✉ & Heath Murray [1] ✉

Genome duplication is essential for the proliferation of cellular life and this process is generally initiated by dedicated replication proteins at chromosome origins. In bacteria, DNA replication is initiated by the ubiquitous DnaA protein, which assembles into an oligomeric complex at the chromosome origin (*oriC*) that engages both double-stranded DNA (dsDNA) and single-stranded DNA (ssDNA) to promote DNA duplex opening. However, the mechanism of DnaA specifically opening a replication origin was unknown. Here we show that *Bacillus subtilis* DnaA^ATP assembles into a continuous oligomer at the site of DNA melting, extending from a dsDNA anchor to engage a single DNA strand. Within this complex, two nucleobases of each ssDNA binding motif (DnaA-trio) are captured within a dinucleotide binding pocket created by adjacent DnaA proteins. These results provide a molecular basis for DnaA specifically engaging the conserved sequence elements within the bacterial chromosome origin basal unwinding system (BUS).

Transmission of genetic material is a requirement for cell proliferation. In most cases DNA replication commences once per cell cycle at loci termed origins, initiating the process of genome duplication. DNA replication requires the anti-parallel strands of the double helix to be separated, enabling templated DNA synthesis by polymerases. Helicases drive the DNA unwinding process by positioning themselves at the leading edge of a replication fork[1]. Across the domains of life, DNA replication initiation proteins containing a conserved AAA+ (*A*TPase *A*ssociated with various cellular *A*ctivities) motif assemble into oligomers on double-stranded DNA (dsDNA) and orchestrate loading of the replicative helicase[2]. Cellular replicative helicases are toroid motor proteins that function by translocating along a single DNA strand (while excluding the opposite strand)[3]. Therefore, a fundamental question is how these helicases are loaded around single-stranded DNA (ssDNA) at a chromosomal origin[4]. Interestingly, it appears bacteria and eukaryotes utilize distinct pathways to achieve helicase loading; bacteria load

helicases directly onto ssDNA strands while eukaryotes initially load helicases around dsDNA before ejecting one of the strands[1]. The unique and essential characteristics of bacterial DNA replication initiation make it an attractive target for drug development.

In bacteria, bidirectional DNA replication typically proceeds from a single chromosome origin (*oriC*) where the ubiquitous master initiation protein DnaA opens the dsDNA to generate the ssDNA substrates for helicase loading (Fig. 1A). DnaA is a multifunctional enzyme composed of four distinct domains that act in concert during DNA replication initiation (Fig. 1B)[5]. Domain IV contains a helix-turn-helix dsDNA binding motif that specifically recognizes a 9 base pair DnaA-box (consensus 5′-TTATCCACA-3′)[6–8]. Domain III is composed of the AAA+ motif that can assemble into an ATP-dependent right-handed helical oligomer[9–11]. Domain III also contains residues that contact the phosphodiester backbone of ssDNA, along with an unidentified ssDNA binding motif that specifically recognizes the trinucleotide DnaA-trio

[1]Centre for Bacterial Cell Biology, Biosciences Institute, Newcastle University, Newcastle Upon Tyne NE2 4AX, UK. [2]Centre for Molecular Cell Biology, School of Biological and Behavioural Sciences, Queen Mary University of London, Newark Street, London E1 2AT, UK. [3]The Francis Crick Institute, 1 Midland Road, London NW1 1AT, UK. ✉e-mail: a.ilangovan@qmul.ac.uk; heath.murray@newcastle.ac.uk

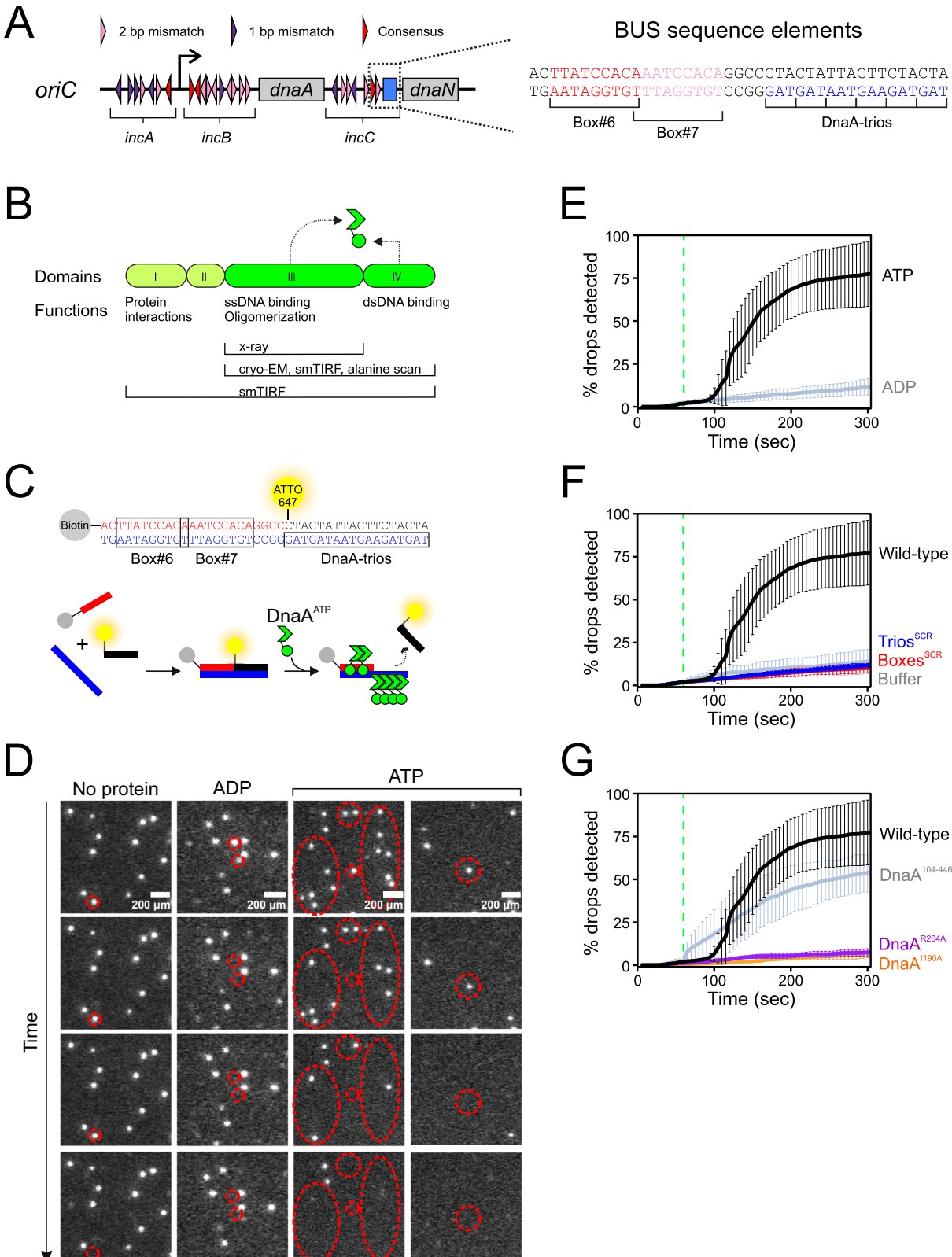

**Fig. 1 | DnaA strand separation events visualised by single molecule TIRF microscopy. A** Schematic representation of the *B. subtilis* replication origin and BUS sequence elements. DnaA-boxes are indicated with triangles. **B** Linear arrangement of DnaA domains I-IV. Locations of critical activities are shown. Endpoints of DnaA proteins used for various experiments are indicated. **C** Schematic describing single molecule TIRF microscopy experiments to investigate BUS activity. BUS sequence elements are indicated by colour. **D** Fields of view from single molecule TIRF experiments performed using different conditions (indicated on top) as a function of time. Red circles indicate spots that disappear over time.

**E** Graph representing the percentage of fluorescent signals with decreased intensity over time in the presence of either ATP or ADP. **F** Graph reporting the percentage of fluorescent signals with decreased intensity over time using wild-type or mutant scaffolds. **G** Graph representing the percentage of fluorescent signals with decreased intensity over time using DnaA variants. DnaA[104–446] lacks domains I-II. DnaA[R264A] is defective in oligomerisation. DnaA[I190A] is defective in ssDNA binding. For all graphs, data shows the average and percentage standard deviation from three independent experiments (source data are provided as a Source Data file).

(consensus 3′-GAT-5′)[12–14]. Domain II tethers domains III-IV to domain I, and domain I acts as an interaction hub that facilitates loading of the replicative helicase[15].

Bacterial replication origins encode information that promotes specific unwinding of the DNA duplex by DnaA[16,17]. However, comparisons of *oriC* regions suggested that they are diverse[18,19]. This heterogeneity hindered the identification of conserved chromosome origin sequence features. Recently, analysis of *oriC* in *Bacillus subtilis* has led to a model for a broadly conserved bacterial chromosome origin *B*asal *U*nwinding *S*ystem (BUS, Fig. 1A)[20]. In this reaction, DnaA binding to DnaA-boxes (dsDNA binding) stimulates the assembly of DnaA[ATP] into an oligomer that engages DnaA-trios (ssDNA binding) to promote DNA strand separation. Because the DNA elements of the BUS are present throughout the bacterial domain[20], this mechanism for DnaA-dependent strand separation is likely ancient.

While genetic and biochemical assays support a model for the BUS where a pair of co-orientated DnaA-boxes recruit DnaA to *oriC* and direct DnaA[ATP] oligomerization onto a proximal set of DnaA-trios[13,20,21], alternative mechanisms for DnaA delivery from DnaA-boxes onto ssDNA have been proposed[10,22,23]. Furthermore, the structure of a DnaA homolog in complex with a non-native ssDNA substrate did not reveal any contacts between the protein and the nucleobases[12]. Therefore, the molecular basis for DnaA promoting specific strand separation at *oriC* is unknown.

In this work, we immobilized DNA substrates and directly visualized strand separation using single-molecule total internal reflective fluorescence (TIRF) microscopy to determine how BUS sequence elements promote specific DnaA oligomer formation. This analysis indicates that DnaA can be loaded from DnaA-boxes directly onto adjacent DnaA-trios. Motivated by this finding, we assembled a model BUS nucleoprotein complex and determined its structure using single-particle cryogenic electron microscopy (cryo-EM). The cryo-EM structure shows that DnaA assembles into a continuous oligomer that engages both dsDNA and ssDNA. This analysis also reveals the formation of dinucleotide binding pockets formed by adjacent DnaA protomers that capture specific nucleobases of the DnaA-trio. Together these studies allow us to propose a model for the BUS promoting specific DNA strand separation at *oriC*.

## Results

### DnaA-boxes promote separation of adjacent DNA strands

Based on reconstituted DNA strand separation assays in vitro using short oligonucleotide scaffolds in solution, two models have been proposed for how DnaA might engage DNA within the BUS complex (Supplementary Fig. 1). One posits that a DnaA[ATP] oligomer assembles on dsDNA using domain IV, and that this nucleoprotein complex engages DnaA-trios located on a separate DNA substrate to promote unwinding (separation in trans)[22,24–26]. Another suggests that DnaA binding to DnaA-boxes acts as a platform for directing a DnaA[ATP] oligomer onto DnaA-trios residing within the contiguous DNA strand (separation *in cis*)[10,12,27].

To test whether *B. subtilis* DnaA can act *in cis* on its native BUS sequence elements, fluorescently labelled DNA scaffolds were immobilized on a glass surface and individual strand separation reactions were visualized directly using single-molecule TIRF microscopy. In this assay, a microfluidic system was used to inject biotin-labelled DNA substrates into a reaction chamber containing a functionalized glass coverslip coated with a streptavidin handle for scaffold capture. A DNA concentration was chosen to maintain an average distance of 15.2 μm (±7.4 μm) between immobilized scaffolds, thereby preventing DnaA from binding two DNA molecules simultaneously (length of DNA substrate ~30 nm). Following scaffold immobilization, the reaction chamber was washed extensively to remove any unbound DNA substrates (i.e. prior to injection of DnaA). DnaA strand separation activity results in the detachment of the fluorescently labelled oligonucleotide

from the scaffold into solution, where it can no longer be detected using TIRF illumination (Fig. 1C).

The addition of DnaA[ATP] to the immobilized DNA scaffolds stimulated a reduction in the number of fluorescent molecules detected (255 drops/330 total) compared to both DnaA[ADP] (48 drops/413 total) and a control reaction lacking protein (44 drops/340 total) (Fig. 1D, E). To ensure that DnaA[ATP] was promoting specific strand separation rather than general scaffold disassembly, two oligonucleotides of the DNA substrate were fluorescently labelled (Supplementary Fig. 2A). The double-labelled DNA scaffold showed that DnaA[ATP] specifically separates the fluorescently labelled oligonucleotide that complements the DnaA-trios (ATTO 647, 194 drops/235 total), while it did not affect the interaction between the remaining oligonucleotides (ATTO 565, 170 drops/413 total; note that ATTO 565 was sensitive to photobleaching) (Supplementary Fig. 2B, C).

Scrambling the DNA sequence of either the DnaA-boxes or the DnaA-trios significantly decreased the number of observed unwinding events (Fig. 1F, Boxes[SCR] 61/609; Trios[SCR] 82/706), indicating that the DnaA-dependent strand separation reaction is *oriC* specific. DnaA variants that lack key residues required for either protein oligomerization[10,28] (DnaA[R264A]; the "arginine finger" 24/353) or ssDNA binding[12] (DnaA[I190A] 23/309) were unable to promote strand separation of wild-type BUS scaffolds (Fig. 1G), consistent with a proposed mechanism for DNA stretching by DnaA[ATP] [12]. Conversely, a DnaA variant lacking domains I-II (DnaA[104–446]) remained competent for DNA strand separation (Fig. 1G, 209/387), showing that domains III-IV are sufficient for this activity. Taken together, the results are consistent with a model in which a DnaA[ATP] oligomer is loaded onto DnaA-trios from the adjacent DnaA-boxes to promote DNA strand separation.

### Cryo-EM structure of the BUS

To understand the molecular basis for specific DnaA strand separation activity, we reconstituted the BUS complex in the presence of ATP using DnaA[104–446] and a tailed DNA scaffold containing a dsDNA region (DnaA-box#6 and DnaA-box#7) and a ssDNA region (DnaA-trios) (Supplementary Fig. 3A). Nucleoprotein complexes were crosslinked with glutaraldehyde and isolated using size exclusion chromatography (Supplementary Fig. 3B, C) before being subjected to single particle cryo-electron microscopy analysis (Supplementary Fig. 3D) to obtain a 3D structure of the BUS complex (Supplementary Fig. 4). The refined map revealed a helical structure at an average resolution of 3.2 Å, in which density corresponding to dsDNA protruding away from the spiral structure could be immediately identified (Supplementary Fig. 5A, B). By reducing the contour level of this map, new density could be seen appearing closer to the density of the DNA double helix, suggesting polypeptide engagement with the DNA (Supplementary Fig. 5A). At much lower sigma contour levels, a separate but continuous density for a second smaller helical structure could also be seen (Supplementary Fig. 5A). At higher thresholds, a separate but continuous density for a second smaller helical structure was evident (Supplementary Fig. 5A). Density between the two helical structures was detected, indicating that they are connected (Supplementary Fig. 5A). Overall, the lack of high-resolution features around the DNA and the smaller helical structure suggested flexibility within the complex or heterogeneity among the population. Hence, the two regions of the overall map were independently determined by local refinement with a mask around the respective regions to obtain their structural information (Supplementary Fig. 4). These maps were then merged to reveal the BUS complex, composed of the spiral region at an average resolution of 3.2 Å, DNA protrusion at a resolution of 5.0 Å and the central region at 5.4 Å (Fig. 2A and Supplementary Fig. 5B–G). A model was built into the combined cryo-EM map using an X-ray crystal structure of the *B. subtilis* AAA+ motif (DnaA[106–346]) (Supplementary Fig. 6A–D), de novo tracing, and homology models, revealing the architecture of the BUS complex (Fig. 2A and Supplementary Movie 1).

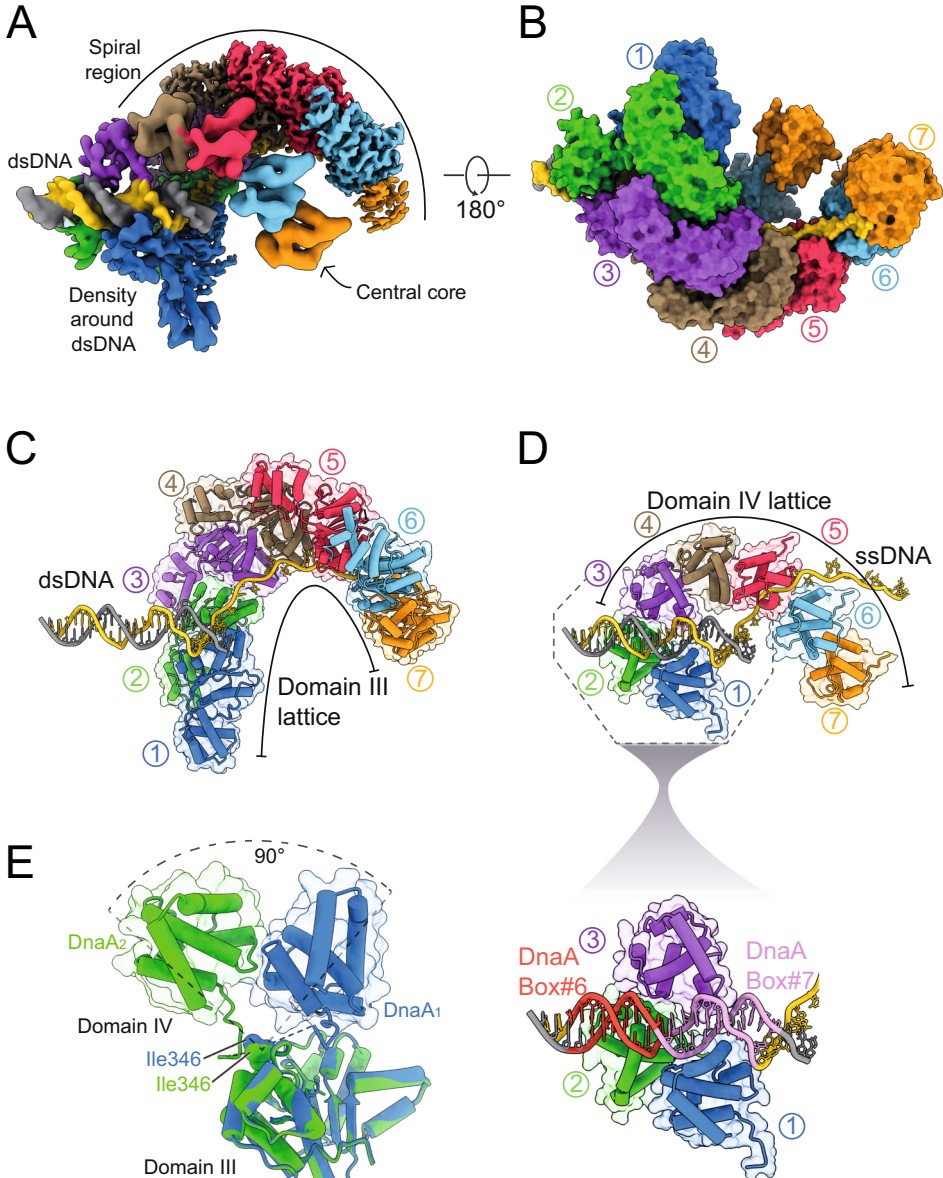

**Fig. 2 | Cryo-EM structure of the BUS complex and dsDNA engagement.**
**A** Composite electron density map of the BUS, resulting from the assembly of maps corresponding to the spiral, central core, and the dsDNA regions contoured at 0.6σ, 0.21σ and 0.27σ respectively. The map is coloured based on seven DnaA protomers (blue, green, purple, brown, pink, cyan and orange respectively), the DnaA-trio containing DNA strand (yellow), and the complementary strand (grey). **B** Surface representation of the BUS complex coloured based on the respective DnaA protomers and DNA strands. **C** Model of the domain III lattice that makes up the spiral region of the map. Both the DNA scaffold and protein are shown in cylinder and stubs representation, with the protein also shown in a transparent surface representation. **D** Model of the domain IV lattice that makes up the central core of the map. Both the DNA scaffold and protein are shown in cylinder and stubs representation with the protein also shown in a transparent surface representation. The enlarged section shows domain IV of $DnaA_1$ and $DnaA_2$ engage with DnaA-box#7 and DnaA-box#6, respectively, while domain IV of $DnaA_3$ engages the posterior minor groove. **E** Structural alignment of the $DnaA_1$ and $DnaA_2$ showing the domain IV of $DnaA_1$ and $DnaA_2$ are 90° apart relative to domain III.

The cryo-EM model reveals seven $DnaA^{ATP}$ protomers (termed $DnaA_{1-7}$) engaging the DNA scaffold as an oligomer (Fig. 2B, Supplementary Fig. 7). Domain III of $DnaA_{1-7}$ forms a continuous lattice of 7 subunits making up the larger helical structure, with adjacent DnaA protomers interacting via the AAA+ motif (Fig. 2B, Supplementary Fig. 7). The DnaA protomers can also be seen engaging the ssDNA region of the scaffold using domain III (Fig. 2B, C). The model shows that the arginine finger residues of DnaA proteins face towards the ssDNA region of the scaffold, away from the DnaA-boxes (Supplementary Fig. 7). This orientation of the DnaA oligomer on the DnaA-trios is compatible with the proposition that DnaA interacts directly with the AAA+ motif of a loader protein to guide helicase deposition[29].

Domain IV of $DnaA_{3-7}$ assembles into a continuous lattice to form the smaller helical structure (Fig. 2D). Here domain IV of $DnaA_{3-7}$ extends away from domain III, in contrast to the crystal structure of DnaA from *Aquifex aeolicus* where domain III and domain IV from adjacent protomers interact[12,27]. Interestingly, *A. aeolicus* DnaA lacks a linker region between domains III and IV (corresponding to *B. subtilis* DnaA residues 333-346), bringing the two protein domains into closer proximity and possibly facilitating the observed interaction.

## DnaA interaction with dsDNA
$DnaA_1$ and $DnaA_2$ use domain IV to interact with the two available double-stranded DnaA-box motifs, engaging with DnaA-box#7 and

DnaA-box#6 respectively (Fig. 2D and Supplementary Movie 2). Both DnaA$_1$ and DnaA$_2$ insert their α16 of domain IV into the major groove of the DNA and make contacts with the bases conferring sequence specificity, as previously reported[8]. Remarkably, DnaA$_3$ could also be observed to interact with dsDNA via its domain IV, where the N-terminal residues of α14 and α17 primarily engage with the backbone phosphates of the minor groove, posterior to where DnaA$_1$ and DnaA$_2$ interact with the DnaA-boxes (Fig. 2D). The structural model suggests that this DnaA$_3$ minor groove interaction may stabilise the domain IV lattice of DnaA$_{3-7}$. No interaction was detected between DnaA$_{4-7}$ and the dsDNA.

DnaA$_1$ and DnaA$_2$ are conformationally distinct in comparison to protomers DnaA$_{3-7}$, with differences arising between the positions of domain IV. Domain IV of DnaA$_1$ is rotated 56° to position it closer to domain III, enabling it to engage DnaA-box#7 (Supplementary Fig. 8A), whereas domain IV of DnaA$_2$ is rotated 37° away from domain III, enabling it to engage DnaA-box#6 (Supplementary Fig. 8B). Thus, domain IV of DnaA has freedom to rotate about 90° to engage with DnaA-box motifs and this flexibility is conferred by the linker region between domains III and IV, notably at the main chain of residue Ile346 (Fig. 2E). The arrangement of DnaA$_1$ and DnaA$_2$ binding dsDNA likely explains the conserved arrangement of tandem DnaA-boxes observed at bacterial BUS sequences[20].

To determine whether the cryo-EM model reflects the physiologically relevant activities of DnaA, we performed a systematic alanine scan of DnaA domains III-IV in *B. subtilis*. Using a parental strain harbouring an ectopic inducible copy of *dnaA*, each residue of domains III-IV in the native *dnaA* gene was individually substituted to encode alanine (Supplementary Fig. 9A). The functionality of the DnaA variants was initially screened by observing cell growth in the absence of ectopic DnaA expression (Supplementary Fig. 9B). Subsequently, the stability of lethal DnaA variants was determined using immunoblotting (Supplementary Fig. 9C), followed by a second growth assay to assess DnaA variant functionality (Supplementary Fig. 10A). Taken together, the alanine screen revealed 37 substitutions that produced a strong growth defect in both growth assays and were expressed at a level similar to wild-type. Highlighting the positions of alanine substitutions that caused severe growth defects onto the cryo-EM model, the importance of residues throughout the initiator specific motif (ISM)[30] is readily apparent, both for protein:protein and protein:DNA interactions (Supplementary Fig. 10B, 11). Critical residues were also identified along the protein:protein interface between domain IIIa and the adjacent domain IIIb, as well as within the recognition motif of the helix-turn-helix used to bind the DnaA-box (Supplementary Fig. 10B, 11). These results support the physiological relevance of the BUS model.

## DnaA:ssDNA interaction and base capture promotes strand separation

The conformation of DnaA$_1$ brings its domains III and IV closer together, placing α5 and α15 about 10 Å apart. This allows the residues on the loop connecting α5 and α6 to interact with the strand encoding DnaA-trios (Fig. 3A). More specifically, the side chain of the essential Lys222 forms a hydrogen bond with the backbone phosphate of G$_{20}$, while the backbone phosphate of G$_{19}$ engages the main chain oxygen of Glu223 (Fig. 3A). These interactions between the DnaA$_1$ domain III and the bases G$_{19}$ and G$_{20}$ (from the strand encoding the DnaA-trios) affects two additional base pairs in the 3' direction, destabilizing the B-form dsDNA (Fig. 3A). Furthermore, helix α3 of the ISM is proximal to the DNA fork and ideally positioned to interrupt the path of DNA progression, splitting the two DNA strands (Fig. 3A). The α3 insertion into dsDNA and Lys222/Glu223 engagement with the backbone phosphate appear to distort the dsDNA structure, either of which might initiate origin unwinding. This is reminiscent of archaeal initiator proteins that bind and alter origin DNA structure using their AAA+ motif[31,32].

DnaA$_1$ makes a third contact with DNA by interacting with G$_{18}$ of DnaA-trio#1, which remarkably is positioned pointing towards the ISM of DnaA$_1$ domain III (Fig. 3B). In this conformation, the backbone phosphate of G$_{18}$ is rotated by 175° in comparison to the backbone phosphate of G$_{19}$. Analogously, A$_{17}$ is stacked with G$_{18}$, although facing the ISM of DnaA$_2$ (Fig. 3B). Thus, G$_{18}$ and A$_{17}$ are captured within a dinucleotide binding pocket created by the ISMs of adjacent DnaA protomers (DnaA$_1$ and DnaA$_2$). In contrast, T$_{16}$ of DnaA-trio#1 is not captured, with the pyrimidine ring pointing the opposite direction of the two previous nucleobases (Fig. 3B). DnaA-trio#2 is engaged by DnaA$_2$ and DnaA$_3$, which generates an analogous nucleotide binding pocket. This pattern continues with each additional DnaA protein, resulting in a total of six DnaA-trios engaging with seven DnaA protomers to generate the domain III lattice (Fig. 3C, D).

Inspection of the DnaA-trio interactions revealed that the two captured bases are stabilized by several hydrogen bond interactions (Fig. 3B and Supplementary Movie 3). The O6 and N2 of guanine make hydrogen bonds with side chains of Arg202 and Glu228, respectively. The N6 of adenine makes a hydrogen bond interaction with the side chain oxygen of Asn187 (Fig. 3B). Two hydrophobic residues, Ile190 and Ile193, appear to flank the third nucleobase of the DnaA-trio, acting to separate it from the adjacent bases (Supplementary Fig. 12). Alanine substitutions at either Asn187, Ile190, Ile193, Arg202, or Glu228 produced a severe growth defect in vivo (Supplementary Fig. 9B, 10A), supporting their key roles within the DnaA oligomer. Sequence alignment of the ISM from DnaA homologs shows that while none of these amino acids are universally conserved, their chemical and physical properties were generally maintained (Fig. 3E, Supplementary Fig. 13A, B).

Nucleobase capture could make a contribution to the mechanism of DNA unwinding by DnaA[33]. To test this hypothesis, we performed strand separation assays using DNA scaffolds with a tetrahydrofuran linkage (THF; abasic mimetic) at the position of the adenine base pair of DnaA-trio#1 (Fig. 3F). Absence of A$_{17}$ from the DnaA-trio significantly impaired strand separation activity, while conversely absence of the complementary thymine nucleoside significantly increased the rate of strand separation (Fig. 3F). Critically, the absence of both nucleobases impaired DNA strand separation (Fig. 3F), showing that unpairing of the adenine within a DnaA-trio stimulates DNA strand separation, consistent with a base capture mechanism promoting opening of *oriC*.

## Molecular basis for DnaA recognizing the DnaA-trio

The cryo-EM model indicates that the amino group of the conserved adenine makes hydrogens bonds with the side chain oxygen of the essential residue Asn187 (Fig. 3B). To assess the functional importance of this interaction, strand separation assays were performed using DNA scaffolds modified at A$_{17}$ (DnaA-trio#1). For these experiments, the fluorescently labelled oligonucleotide annealed to the DnaA-trios contained a THF linkage (abasic site) at the position complementing A$_{17}$ (Fig. 4A), thereby allowing substitution of the nucleobase without the complications of distinct hydrogen bonding interactions (Fig. 4B).

While substitution of A$_{17}$ with either guanine, thymine or uracil decreased the rate of strand separation, substitution with the pyrimidine cytosine resulted in a separation rate similar to that of the native adenine (Fig. 4C). Interestingly, both adenine and cytosine nucleobases contain a primary amine group that acts as an electron donor (Fig. 4B). To investigate the role of this functional group in the DnaA-dependent strand separation reaction, several modified nucleobases were tested (adenine was replaced by hypoxanthine, cytosine was replaced by 5-methyl-isocytosine; guanine was replaced by isoguanine) (Fig. 4B). Hypoxanthine and isocytosine, which both lack the primary amine group, decreased the rate of strand separation compared to adenine and cytosine, respectively (Fig. 4C). Strikingly

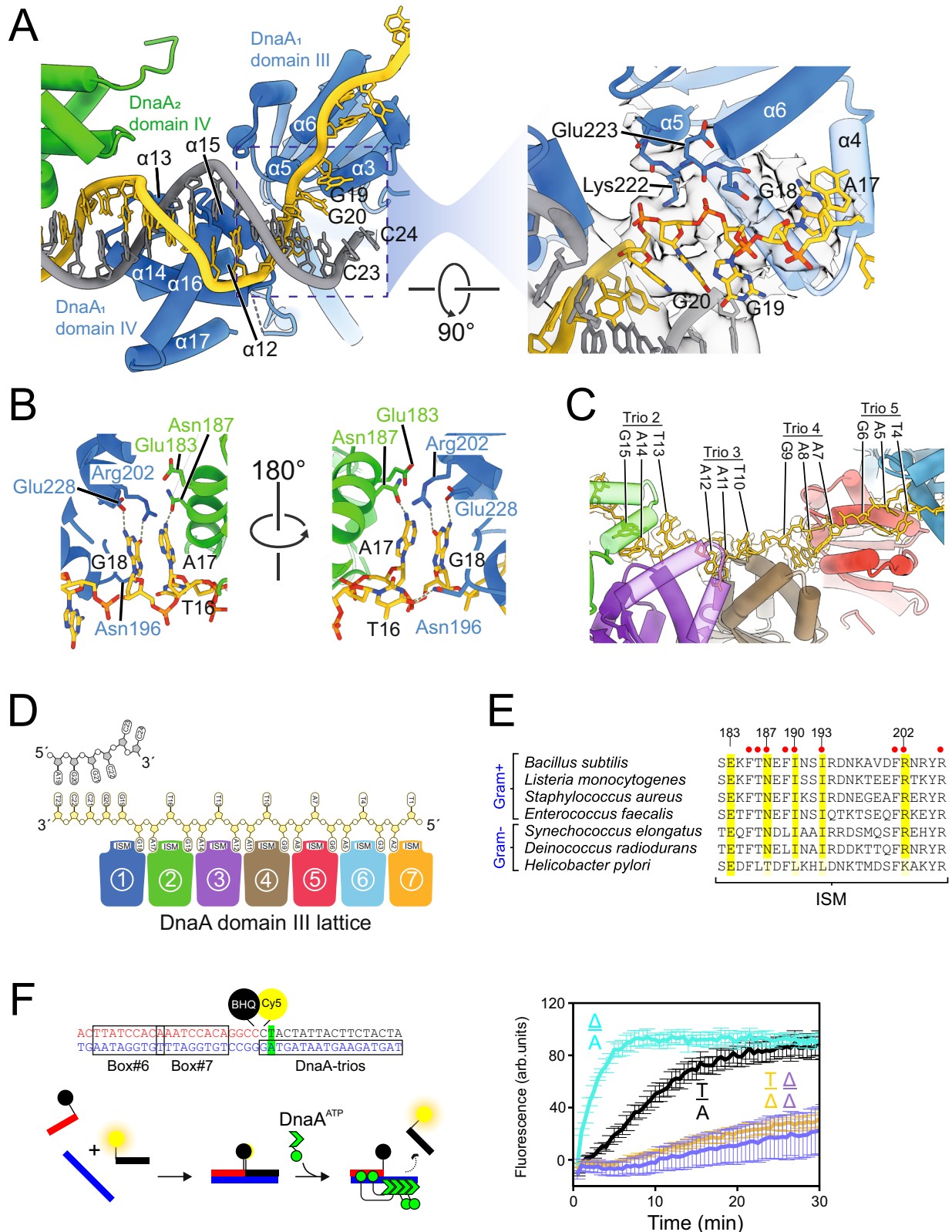

isoguanine, which contains a functional amine group, increased the rate of unwinding compared to guanine (Fig. 4C).

To assess the role of this primary amine in vivo, the essential DnaA-trios (#1-3)[13] were mutated such that the central adenine of each motif was changed to the three alternative nucleobases. Here we utilized a strain in which DNA replication can initiate from a heterologous plasmid origin (*oriN*) integrated into the chromosome (Fig. 4D)[13]. The

activity of *oriN* requires its cognate initiator protein RepN, whose expression is dependent upon the inducer IPTG. A clear pattern was observed, with the cytosine substitutions supporting faster colony growth compared to either guanine or thymine substitutions (Fig. 4E). Marker frequency analysis confirmed that the strain with cytosine substitutions initiated DNA replication more frequently than strains with either guanine or thymine substitutions (Fig. 4F). Taken together,

**Fig. 3 | Two bases of the DnaA-trio are flipped into the DnaA oligomer. A** Model showing engagement of α3 with the DNA scaffold and the loss of pairing between for G19:C24 and G20:C23. The focused image with a 90° rotated view shows residues Lys222 and Glu223 engaging with the phosphate groups of $G_{20}$ and $G_{19}$, respectively, together with the map density contoured at 0.6σ (spiral region map) and 0.34σ (dsDNA region map). **B** A focused view of the DnaA-trio#1 and DnaA-trio#3 interaction with DnaA domain III. Within the bipartite pocket $G_{18}$ makes hydrogen bond interactions with Arg202 and Glu228 of $DnaA_1$, while $A_{17}/A_{11}$ make hydrogen bond interactions with Asn187 and Glu183 of $DnaA_2$ and $DnaA_4$, respectively. Outside the bipartite binding pocket $T_{16}/T_{10}$ makes a hydrogen bond interaction with Asn196 of $DnaA_1$ and $DnaA_3$, respectively. **C** Interaction between the DnaA protomers and ssDNA, showing the two captured and one uncaptured base of DnaA-trio#2, #3, #4 and #5 with the extracted map around the bases contoured at 0.6σ. **D** Schematic showing the DnaA-trio strand engaging with DnaA protomers 1–7, where the first two bases of the DnaA-trio insert into the bipartite dinucleotide binding pocket while the third faces away. **E** Alignment of residues around the DnaA ISM involved in binding DnaA-trios (numbered for *B. subtilis*). Red dots indicate that alanine substitution in vivo is lethal. These DnaA homologs were previously shown to form a functional BUS with their cognate *oriC* sequences[20]. **F** Schematic representation of the BHQ strand separation assay. BHQ-SSA performed using probes containing abasic sites (nucleotide substituted with THF) in the central position of the first DnaA-trio (base pair shown in green). Positions of bases on the top and bottom strand are indicated; Δ denotes an abasic site. Data shows the average and percentage standard deviation from three independent experiments (source data are provided as a Source Data file).

these results support a model in which an amino group on the middle base of the DnaA-trio contacts the essential residue Asn187, thus providing a molecular basis for the DnaA oligomer binding ssDNA specifically at *oriC*.

While the middle adenine of the DnaA-trio motif is conserved in *B. subtilis*, not all DnaA-trios contain an adjacent guanine captured within the dinucleotide pocket. To investigate whether the unique 3′-AA-5′ dinucleotide of DnaA-trio#3 is functionally important, the sequence was replaced (using the *oriN* system described above) and the strains were characterized. No significant defect in either cell viability or the frequency of DNA replication initiation was detected when DnaA-trio#3 was altered (Supplementary Fig. 14A, B). Interestingly, this analysis showed that while having all six DnaA-trios encoding 3′-GA-5′ is functional, a mutated origin with all six DnaA-trios encoding 3′-AA-5′ displayed a defect in both growth and the frequency of DNA replication initiation (Supplementary Fig. 14A, B). These results suggest that while some variation in DnaA-trio sequence motif is permitted, the dinucleotide pair 3′-GA-5′ is the preferred species, likely due to the specific interactions with the DnaA oligomer. Moreover, while a mutant origin with all DnaA-trios encoding adenine at the first position has a lower predicted melting temperature, it is the wild-type motif that achieves the highest rate of DNA replication initiation. These results support the model that efficient nucleobase capture is a critical event that promotes BUS activity.

## Discussion

Based on the data presented, we propose that the *B. subtilis* BUS promotes the capture of specific nucleobases within the DnaA oligomer at *oriC*. The arrangement of two co-orientated DnaA-boxes adjacent to the DnaA-trios allows a pair of DnaA proteins to both localize at *oriC* (dsDNA binding) and assemble their AAA+ motifs to construct a dinucleotide pocket for nucleobase capture (ssDNA binding). Additional DnaA monomers can then be added to this platform to assemble the continuous right-handed helical protein oligomer (Fig. 2). This model is consistent with the observation that DnaA strand separation on BUS scaffolds occurs *in cis* (Fig. 1, Supplementary Fig. 1).

DnaA-box#6 is both composed of the consensus sequence which binds DnaA with the highest affinity[34] and the most critical element of the *B. subtilis* origin in vivo and in vitro[13,20,21]. We propose that $DnaA_2$ is bound stably to DnaA-box#6 and that $DnaA_1$ binding to DnaA-box#7 would likely occur subsequently, thereby allowing the first dinucleotide pocket to be formed. While $DnaA_3$ may contact dsDNA non-specifically, $DnaA_{4-7}$ do not appear to contact the dsDNA region of the scaffolds used in this study (Fig. 2A–D). It will be necessary to extend scaffolds with upstream dsDNA to determine how these influence BUS structure. We note that if $DnaA_{4-7}$ were to contact dsDNA, this would require that the domain IV lattice be repositioned to allow engagement of the major groove.

We envision that the captured nucleobases of the DnaA-trios could be either flipped out from one strand of duplex DNA to drive

open complex formation or captured following initial base pair destabilization. In both cases, nucleobase capture would exclude pairing with the complementary strand, thereby stabilizing the open complex to allow helicase loading around ssDNA.

Prior to nucleobase capture, it appears that destabilisation of the DNA duplex by DnaA could arise from stretching the phosphodiester bonds of the strand containing DnaA-trios. Overlaying the ssDNA from available DnaA nucleoprotein complexes (*B. subtilis* cryo-EM and *Aquifex aeolicus* X-ray[12] structures, with and without base flipping, respectively) shows that the backbones adopt the same extended conformation (Fig. 4G). We hypothesize that DnaA could either assemble into an oligomer along the ssDNA backbone and stretch the substrate through multiple contact points to liberate the bases for capture, or sequentially proceed through a cycle of steps in which the addition of each DnaA protomer would stretch the backbone and capture the flipped nucleotides before the next DnaA protomer can bind.

Phylogenetic analysis supports the hypothesis that the BUS is broadly conserved (Fig. 3E, Supplementary Fig. 13A, B)[20]. However, the observation that cytosine can functionally replace the central adenine within DnaA-trios (Fig. 4A–F) raises the possibility that DnaA homologs could recognize a range of sequences within ssDNA to promote specific base capture. Moreover, the observations that DnaA-box#7 is not essential for either origin activity in vivo[13] or strand separation in vitro[20], and that the region linking DnaA domain III and domain IV is flexible (Fig. 2E)[10], suggests that BUS architecture could vary from that observed in *B. subtilis*. This plasticity could help explain the apparent differences observed between origins throughout the bacterial domain[20].

While the proposed model provides a new framework for investigating the opening of replication origins in diverse bacterial species, many fundamental questions remain regarding bacterial DNA replication initiation. For examples, we do not know how changes in local supercoiling[10] or direct interactions with the ISM (Fig. 3A)[31,32] contribute to open complex formation; we do not understand how dual helicase loading is orchestrated to achieve bidirectional DNA synthesis from *oriC*; we still have yet to discover how DNA replication initiation is coordinated with other essential cell cycle activities.

It is timely to note that the bacterial DNA replication machinery is an attractive subject for drug development. The process is essential for proliferation, the bacterial proteins are distinct from functional eukaryotic analogues, and no current antibiotics in clinical use target this pathway[35,36]. Obtaining a clearer view of the bacterial DNA replication initiation mechanism will facilitate the development of small molecule inhibitors to disrupt this vital cellular process.

## Methods
### Media and chemicals
Nutrient agar (NA; Oxoid) was used for routine selection and maintenance of *E. coli* and *B. subtilis* strains. Supplements for *B. subtilis* cell culture were added as required: 5 µg/ml chloramphenicol, 50 µg/ml spectinomycin, 10 µg/ml tetracycline, 1 µg/ml erythromycin in

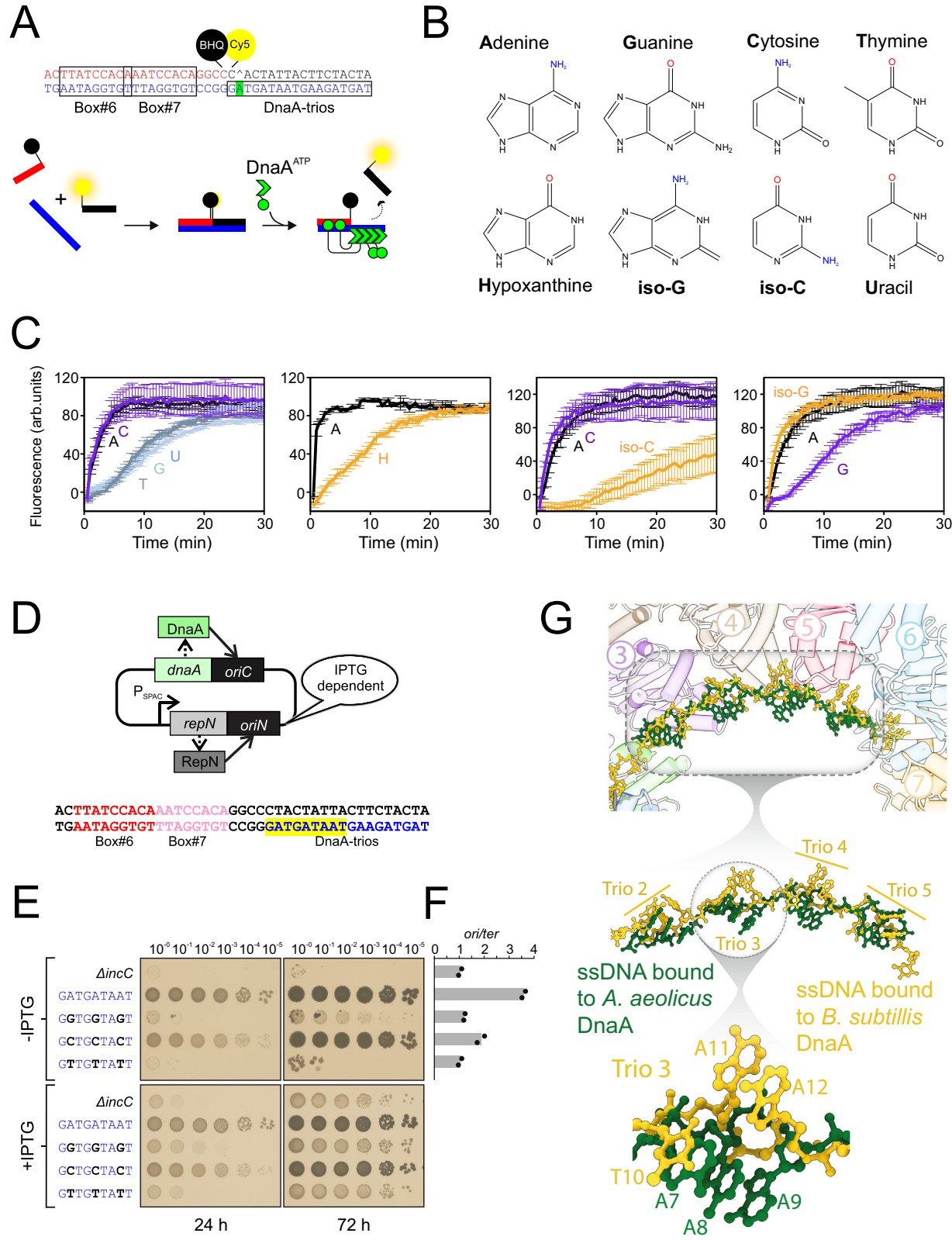

conjunction with 25 µg/ml lincomycin, 0.016% w/v X-gal, 1% w/v xylose, 0.1 mM IPTG. For protein expression, cells were grown using 2X YT medium (16 g tryptone, 10 g yeast extract, 5 g NaCl for 1 L)[37]. Supplements were added as required: 1 mM IPTG, 100 µg/ml ampicillin. Plasmid extractions were performed using Qiagen miniprep kits. Unless otherwise stated, all chemicals and reagents were obtained from Sigma-Aldrich.

**Glass coverslip functionalization**
Glass coverslips (22 × 50 mm; VWR) were functionalized in clean staining jars and all solutions were pre-filtered. Glass surfaces were sonicated four times, 30 min each, in a water bath (GT SONIC-D6). After each sonication step, coverslips were washed three times with milliQ water. Solutions for sonication were: EtOH (100%), KOH (1 M), EtOH (100%), KOH (1 M). After the final sonication step, jars were

**Fig. 4 | The amine group on adenine mediates specificity for DnaA-trio recognition by DnaA. A** Schematic representation of BHQ strand separation assay. The fluorescently labelled oligonucleotide contains an abasic site complementary to the central adenine (base shown in green). **B** Structures of nucleobases used to replace adenine. **C** Strand separation assay performed with nucleotide substitutions at the central adenine of the first DnaA-trio, in the context of an abasic site at the complementary position. Positions of mutations and structure of modified bases are indicated. Data shows the average and percentage standard deviation from three independent experiments (source data are provided as a Source Data file). **D** Schematic of the inducible *repN/oriN* system used to bypass mutations affecting *oriC* activity in *B. subtilis*. Replication via *oriN* is turned on and off in the presence and absence of IPTG, respectively. **E** Spot titre analysis with strains that have replaced the central adenine of DnaA-trio#1-3. The presence or absence of IPTG indicates the induction state of the *repN/oriN* system. Wild-type (FDS682), ΔincC (FDS404), A→G (FDS688), A→C (FDS686), A→T (FDS691). **F** Marker frequency analysis of strains shown in panel F. Data shows the average and individual data points from two independent experiments (source data are provided as a Source Data file). **G** Comparison of ssDNA substrates from DnaA nucleoprotein complexes. The synthetic polyA ssDNA from *A. aeolicus* (shown in green) (PDB 3R8F) was superimposed onto the BUS ssDNA (shown in yellow). Insets provide greater detail of the distinct conformations adopted by the two ssDNA substrates.

washed five times with milliQ water, filled with KOH (1 M), and left to stand overnight. Jars were washed three times with milliQ water and then three times with acetone, prior to drying the coverslips using an $N_2$ gun. A plasma cleaner was used to clean the coverslips (0.5 torr, HIGH setting for 5 min; Harrick Plasma cleaner 115 V) before placing them back in staining jars filled with acetone. To silanize the glass surfaces, coverslips were incubated in a 2% solution of (3-Aminopropyl)triethoxysilane (APTES, Sigma-Aldrich) in acetone with agitation for two minutes. The reaction was stopped by pouring water directly into the jars. Three $0.2 \times 4$ mm chambers were prepared using a double-sided adhesive (TESA 4965) and applied on the silanized glasses after drying them with an $N_2$ gun. A 46 μM solution of PEG/biotin-PEG (in a 56:1 ratio; Laysan Bio Inc.) dissolved in fresh 100 mM $Na_2CO_3$ was applied to each chamber and left to bind for 3 h. Chambers where then rinsed thoroughly with water, dried with an $N_2$ gun, and stored in an anaerobic atmosphere.

### Microfluidic flow chamber preparation
Six holes, three each side positioned to fit the chamber of the gasket, were drilled into a microscope slide (VWR) using a DREMEL 3000 rotatory tool and a 1 mm diamond-coated drill tip (UKAM Industrial Superhard Tools). The previously prepared gasket coverslips were applied, aligning each chamber to the holes (one influx and one efflux). Two polypropylene tubes were allocated in each one of those (BD intradermic, efflux: $0.76 \times 1.22$ mm influx: $0.38 \times 1.09$ mm) and fixed in place with fast dry epoxy resin (Araldite). Prior to use, each chamber was filled for 30 min with a 1 mg/ml solution of Streptavidin (NEB) and then passivated with a blocking solution (20 mM Tris-HCl pH 7.5, 50 mM NaCl, 0,5 mg/ml BSA, 1% Tween).

### Single-molecule imaging
All the solutions were flowed into the chamber using an infusion pump (WPI, AL-1000) with a flow rate of 30 μl/min. Before imaging, chambers were washed thoroughly with Strand Separation Assay (SSA) buffer (30 mM HEPES pH 7.6, 100 mM potassium glutamate, 10 mM magnesium acetate), then DNA (1 fM) was applied for 1 min. The unbound molecules were washed away with SSA buffer followed by the same solution containing oxygen scavengers (10 mg/ml glucose oxidase, 0.4 mg/ml catalase, 1 mM Trolox, 0.8 mg/ml of D-glucose) to stabilize the fluorophores. Images were taken using a Nikon Ti eclipse inverted microscope housing an oil immersion objective (CFI SR HP Apo TIRF 100XC Oil). The fluorophore ATTO[647] was excited using a 300 mW fibre laser at 647 nm wavelength (MPB communications Inc.) with a power intensity of 583.1 μW (±2.405 μW) at the sample and 500 ms exposure time. Fluorophore ATTO[565] was excited using a 150 mW laser at 561 nm wavelength (Coherent) with a power intensity of 1444 μW (±0.853 μW) at the sample and 500 ms exposure time. The laser power reaching the sample was measured with a PM100 power meter (Thorlabs). The emitted fluorescence was filtered using a 405/488/561/647 nm Laser Quad Band Set (Chroma) and the signal was detected and recorded using an EMCCD iXon X3 897 camera (ANDOR technology). Images were captured every 5 s for 10 min using Nikon NIS Elements software (v5.42.01). Protein solutions (30 μl) were diluted to a final concentration of 50 nM in SSA buffer containing oxygen scavengers and flowed in to initiate the reaction.

### Microscopy data analysis
TIRF timelapse images were processed automatically within Fiji (ImageJ 1.53c)[38] for drift-correction, then maxima were assessed to define initial molecule location regions of interest. The regions were measured over time to obtain fluorescence intensity relative to the initial intensity, with drop detection triggered if fluorescence intensity drops below 75% of the initial intensity for 5 sequential frames. A fully annotated ImageJ Macro is available on GitHub (https://github.com/GMerces/DisappearingSpots). Graphs were prepared using SigmaPlot (v14).

### DNA scaffolds
DNA scaffolds were prepared by adding each oligonucleotide (10 μM) to a 20 μl volume containing 30 mM HEPES-KOH (pH 8), 100 mM potassium acetate and 5 mM magnesium acetate. Mixed oligonucleotides were heated to 95 °C for 5 min and then cooled 1 °C/min to 20 °C in a PCR machine before being stored at 4 °C. Assembled scaffolds were diluted to 1 μM and stored at −20 °C.

DNA scaffolds were imaged with a Typhoon FLA 9500 laser scanner (GE Healthcare). Cy5 labelled oligonucleotides were excited at 400 V with excitation at 635 nm and emission filter LPR (665LP). Cy3 labelled oligonucleotides were excited at 400 V with excitation at 532 nm and emission filter LPG (575LP). Images were processed using Fiji subtracting the background and adjusting the overall contrast[38]. All experiments were independently performed at least thrice and representative data is shown. The melting temperature of each probe was measured by preparing a 20 μl reaction containing 0.4 μM of probe in SSA buffer. A melting curve determination experiment from 20 to 95 °C was then performed on the reaction on a Rotor-Gene Q qPCR machine (Qiagen) with 563 excitation filter and 610 high-pass emission filter (gain:10). The measured melting temperature of each DNA scaffold is reported in Supplementary Data 1 (average and standard deviation from three replicates).

### Black Hole Quencher DNA strand separation assay (BHQ-SSA)
Each DNA scaffold containing one oligonucleotide with BHQ2, one labelled with Cy5 and one unlabelled (12.5 nM final concentration) was diluted in SSA buffer. Reactions were performed using a flat-bottom black polystyrene 96-well plate (Costar #CLS3694) in a BMG Clariostar platereader (software v5.20). All reactions were made in triplicate, with fluorescence output measured every 30 s for a total of 75 min, allowing the first point to be measured without the protein and then adding DnaA at a final concentration of 650 nM. For all reactions a negative control without protein was performed in triplicate (background). At each timepoint the average background value was subtracted from the experimental value, thus reporting the specific DnaA activity on a single substrate. All reactions were prepared on ice to ensure the stability of the DNA probe, then allowed to equilibrate to the reaction temperature (20 °C) before starting the measurement. Graphs were prepared using SigmaPlot (v14).

## DnaA protein purification

Plasmids encoding for $His_{14}$-SUMO-DnaA and $His_{14}$-SUMO-$DnaA^{104-446}$ proteins were transformed into BL21(DE3)-pLysS. Strains were grown in 2X YT medium at 37 °C and at $A_{600}$ 0.4, 1 mM IPTG was added and cultures were incubated for 4 h at 30 °C. Cells were harvested by centrifugation at 7000 g for 20 min, resuspended in 40 ml of $Ni^{2+}$ Binding Buffer (30 mM HEPES [pH 7.6], 250 mM potassium glutamate, 10 mM magnesium acetate, 30 mM imidazole) containing 1 EDTA-free protease inhibitor tablet (Roche #37378900) and flash frozen in liquid nitrogen. Cell pellet suspensions were thawed and incubated with 0.5 mg/ml lysozyme with agitation at 4 °C for 1 h. Cells were disrupted by sonication (Fisherbrand 505, 60 min of 20 sec ON/OFF cycles at 30% power with a ¼ inch tip in an ice bath). To remove cell debris the lysate was centrifuged at 24,000 g for 30 min at 4 °C, then passed through a 0.2 μm filter for further clarification. All the further purification steps were performed at 4 °C using Fast Protein Liquid Chromatography (FPLC) with a flow rate of 1 ml/min.

The clarified lysate was applied to a 1 ml HisTrap HP column (GE), washed with 10 ml $Ni^{2+}$ High Salt Wash Buffer (30 mM HEPES [pH 7.6], 1 M potassium glutamate, 10 mM magnesium acetate, 30 mM imidazole) and 10 ml of 10% $Ni^{2+}$ Elution Buffer (30 mM HEPES [pH 7.6], 250 mM potassium glutamate, 10 mM magnesium acetate, 1 M imidazole). Proteins were eluted with a 10 ml linear gradient (10-100%) of $Ni^{2+}$ Elution Buffer. Fractions containing the protein were applied to a 1 ml HiTrap Heparin HP affinity column (GE) equilibrated in H Binding Buffer (30 mM HEPES [pH 7.6], 100 mM potassium glutamate, 10 mM magnesium acetate). Proteins were eluted with a 20 ml linear gradient (20–100%) of H Elution Buffer (30 mM HEPES [pH 7.6], 1 M potassium glutamate, 10 mM magnesium acetate). Fractions containing DnaA (usually 3 ml total) were pooled and digested overnight with 10 μl of 10 mg/ml $His_{14}$-TEV-SUMO protease[39].

The digested reaction product was applied to a 1 ml HisTrap HP column to capture uncleaved protein, $His_{14}$-SUMO-tag and $His_{14}$-TEV-SUMO protease. Cleaved DnaA proteins were collected in the flow-through and their purity was confirmed on SDS-PAGE. Glycerol was added (20% final) and protein aliquots were flash frozen in liquid nitrogen before being stored at −80 °C.

$His_{14}$-SUMO-$DnaA^{109-346}$ was purified using the protocol described above, with the following modifications. The HiTrap Heparin column was omitted, and after SUMO digestion $DnaA^{109-346}$ protein was dialysed into buffer containing 25 mM HEPES-KOH [pH 7.6], 250 mM potassium glutamate, 10 mM magnesium acetate and 1 mM TCEP overnight. After removal of the $His_{14}$-TEV-SUMO protease and $His_{14}$-SUMO-tag, $DnaA^{109-346}$ protein was dialysed into buffer containing 25 mM HEPES-KOH [pH 7.6], 250 mM potassium glutamate, and 2 mM EDTA to remove bound nucleotides. To subsequently remove EDTA, $DnaA^{109-346}$ protein was dialysed into buffer containing 25 mM HEPES-KOH [pH 7.6] and 250 mM potassium glutamate. After dialysis, the buffer was supplemented with 10 mM MgOAc and 1 mM ADP (final concentrations). Finally, $DnaA^{109-346}$ protein was purified by size exclusion chromatography (HiLoad 26/600 Superdex 75) in buffer containing 20 mM HEPES [pH 7.6] and 100 mM NaCl before adjusting the concentrated to 16 mg/ml (Amicon Ultra-15 Centrifugal Filter Unit) and freezing in liquid nitrogen.

## DnaA protein crystallisation

$DnaA^{106-346}$ was dialysed into 20 mM bis-tris propane [pH 8], 100 mM NaCl, 10 mM MgOAc. The protein concentration for the screen was 11 mg/ml. Crystals growing from one point appeared in the condition 0.03 M sodium nitrate, 0.03 M sodium phosphate dibasic, 0.03 M ammonium sulfate, 0.1 M imidazole, MES [pH 6], 20% v/v ethylene glycol, 10% w/v PEG 8000 (Morpheus, C2) and were used to prepare seeds. For AMP-PNP co-crystallisation, protein was dialysed with 2 mM EDTA, followed by addition of 1 mM AMP-PNP and 10 mM MgOAc. Protein concentration for the screening was 8.8 mg/ml and

microseeding was used to obtain crystals. Some small split crystals grew and only one large needle in Morpheus H3 condition (0.1 M DL-glutamic acid monohydrate; 0.1 M DL-alanine; 0.1 M glycine; 0.1 M DL-lysine monohydrochloride; 0.1 M DL-serine, 0.1 M imidazole, MES [pH 6.5], 20% glycerol, 10% PEG 4000). The dataset was collected at Diamond I24 beamline at 100 K using synchrotron radiation. Data were processed with AutoPROC[40] and output from STARANISO[41] with diffraction limits 2.380 Å, 2.380 Å and 2.647 Å was used to solve the structure with Phaser[42] using a Consurf[43,44] predicted model for molecular replacement. The structure was refined with Refmac[45] (Supplementary Table 1). The Ramachandran plot shows 94.59 % of residues in favourable region, 5.41 % in allowed region, and none in disallowed region.

## Sample preparation for single particle cryo-EM analysis

The BUS nucleoprotein complex was prepared by incubating 4 μM of partially ssDNA scaffold with 40 μM $DnaA^{104-446}$ monomer in a buffer containing 10 mM HEPES [pH 7.5], 100 mM potassium glutamate, 1 mM magnesium acetate and 2 μM ATP at 37 °C for 10 min before adding 0.1% glutaraldehyde and incubating for an additional 60 min. The reaction was stopped by the addition of 100 mM Tris HCl [pH 7.5] and then subjected to size exclusion chromatography (SEC) using a Superdex 200 10/300 GL Increase column to separate different species. Peak fractions corresponding to a multimeric complex were collected and subjected to cryo-EM analysis.

For BUS structure determination by cryo-EM, the crosslinked sample was diluted 10-fold with a buffer containing 20 mM HEPES [pH 7.5], 100 mM NaCl and 1 mM DDM to make a final concentration of 0.25 mg/ml. Quantifoil Au 2/2 200 grids were plasma cleaned using the Harrick Plasma cleaner for 15 sec at low and 60 sec at high plasma setting for cleaning and preparing the grid prior to cryo grid preparation. Samples were vitrified in liquid ethane using a Leica automatic plunge freezer EM GP2 with a 2.4–2.8 s blot time under 85 % relative humidity at 4 °C. The frozen grids were checked for sample concentration and ice thickness using the Jeol 2100 plus microscope. The grids with ideal protein concentration and ice thickness were used for data acquisition at National cryo-EM facility at eBIC, located within the Diamond light source using FEI Titan Krios operating at 300 kV. Images were collected automatically using EPU software (FEI) on a Gatan K3 detector in counting mode with a pixel size of 1.072 Å. A total of 17132 movies were recorded with a nominal defocus range of approximately −0.1 to −3.0 μm. Each image consisted of a movie stack of 50 frames with a total dose of 48.3 $e^-$/Å$^2$ over 3.9 s corresponding to a dose rate of 15 $e^-$/px/sec.

## Cryo-EM image processing and reconstruction

The movie stacks were aligned and summed using patch MotionCor2[46] using Relion[47] via the on-the-fly processing pipeline at eBIC. The motion-corrected micrographs were imported into cryoSPARC (v3.1.0) and micrograph CTF was estimated using the patch CTF estimation function[48]. After screening the micrographs for good Thon Rings and ideal ice thickness, 16616 micrographs remained in the dataset for further processing. The blob picking tool was used to pick an initial set of 300,000 particles from a subset of 500 micrographs that was classified to generate the initial template for subsequent rounds of picking. Once the right templates and picking parameters were obtained, a total of 4,970,535 particles were picked from the entire dataset and extracted using a box size of 320 × 320 with 2-fold binning. Low-quality particles were removed after several rounds of 2D classification in cryoSPARC and Relion, resulting in a stack of 1,192,718 particles. These particles were classified into 5 ab initio classes while simultaneously generating 5 novel volumes. This total stack of 1,192,718 unbinned particles was re-extracted and used for 3D classification by heterogenous refinement using the 5 ab initio volumes as templates. This heterogenous refinement/classification job segregated the particles into 5 classes constituting 318,478 particles (class 1),

227,717 particles (class 2), 233,260 particles (class 3), 187,855 particles (class 4) and 225,408 particles (class 5). Of this classification only class 1 with 318,478 was subjected to non-uniform refinement which yielded a map where the spiral region and DNA were visible while additional density for the central core and peptide contacting DNA showed up at lower sigma threshold. The region constituting the dsDNA and DnaA$_1$ and DnaA$_2$ were processed separately using local refinement in cryoSPARC. The entire set of 1,192,718 particles was used to obtain a high-resolution map of this region after signal subtraction, which yielded a map at 5.0 Å (FSC = 0.143). For the domain IV lattice, particles corresponding to classes 1, 3 and 5 (777,146 particles) were used. These particles were subjected to signal subtraction and local refinement which yielded a map at 5.4 Å (FSC = 0.143). A composite BUS complex map was generated using the maps from these three regions. Statistics for data collection and 3D refinement are included in Supplementary Table 2.

### Model building and refinement

Model building started with identifying the three segments of the maps. The spiral part of the map could immediately be identified as the domain III lattice. A crystal structure of the *B. subtilis* DnaA domain III was used to rigid body fit into the density. A single DnaA domain was isolated from the crystal structure and placed into the density using chimera thus building the entire domain III lattice. The ATP molecule was manually built using Coot and the model was subjected to several iterative rounds of real space refinement using Phenix[49] and progress in refinement was tracked using Ramachandran plot and Molprobity[50]. For the central core a homology model of the domain IV was obtained by using the *B. subtilis* DnaA domain IV sequence in Phyre2[51] that was then used to rigid body fit into the density. This made the domain IV lattice constituted by domain IVs of DnaA$_{3-7}$. For the map of double-stranded DNA region, a double-stranded B-form DNA was manually built and with the sequence using Coot. The DnaA domain IV homology model was used to rigid body fit into the density around the dsDNA. This model was subjected to several rounds of real space refinement using Phenix and modification in Coot while progress was monitored using Ramachandran plot and Molprobity. A final combined model of the BUS complex was generated by merging the different coordinate files using the composite BUS complex map. A single BUS complex coordinate file has been submitted to the Protein Data Bank (PDB) with entry code 8BTG.

### Immunoblotting

Fixed samples were resolved on a NuPAGE Novex 4–12% Tris-Acetate Gel (Thermo Fisher Scientific) then transferred to a PVDF membrane using Turbo-Blot transfer apparatus and Trans-Blot TurboTM Midi PVDF Transfer Packs (Bio-Rad). The membrane was blocked with PBS + 5% milk for 60 min at room temperature then incubated with a PBS + 1% milk solution containing either a 1:2000 custom polyclonal anti-DnaA antibody (Eurogentec) or a 1:10,000 dilution of custom polyclonal anti-FtsZ antibody (Eurogentec) at 4 °C overnight. Antibody specificity has been confirmed by performing immunoblots using strains lacking either DnaA[21] or FtsZ[52]. The membrane was washed three times with PBS + 0.05% Tween-20 and then incubated with PBS + 1% milk solution containing a 1:10,000 anti-rabbit horse-radish peroxidase conjugated secondary antibody (A0545, Sigma-Aldrich). The membrane was then washed three times with PBS + 0.05% Tween-20 and then incubated for 5 min with Pierce ECL Plus substrate (Thermo Scientific). Chemiluminescence was detected using an ImageQuant LAS 4000 imager (Amersham). Images were processed using Fiji[38].

### Phenotype analysis of dnaA alanine substitution mutants

Strains were grown at 37 °C either in LB with or without xylose (1%) for plate reader assays or on NA plates either with or without xylose (1%)

for 24 h for spot-titre assays. For growth in liquid media, culture absorbance (600 nm) was monitored using a Tecan Sunrise plate reader running Magellan software (v7.2). For growth on solid media, plate images were captured using a Perfection V800 Photo scanner (Epson). All experiments were independently performed at least twice and representative data are shown.

### Phenotype analysis of origin mutants via spot titre assays

Strains were grown in test tubes overnight to saturation at 37 °C in LB supplemented with 0.1 mM IPTG to maintain replication initiation from *oriN*. The cell cultures were washed once by centrifugation and resuspended in LB lacking IPTG, set to a starting A$_{600}$ of 0.5 and then serially diluted 1:10 in LB lacking IPTG. 5 μL of each dilution were spotted on nutrient agar plates with or without 0.1 mM IPTG, respectively. Plate images were captured using a Perfection V800 Photo scanner (Epson). All experiments were independently performed at least twice and representative data are shown.

### Marker frequency analysis

Relative amounts of origin (*ori*) and terminus (*ter*) DNA were determined by qPCR. Strains were grown in flasks to an A$_{600}$ of 0.1-0.4 at 37 °C in LB supplemented with 0.1 mM IPTG to maintain replication initiation from *oriN*. The cell cultures were washed twice by centrifugation and resuspension in LB lacking IPTG, set to a starting OD$_{600}$ of 0.05 in LB lacking IPTG and grown to an OD$_{600}$ between 0.3 and 0.45 at 37 °C. Five hundred microliters of culture were mixed with sodium azide to a final concentration of 0.05 % w/v to arrest growth and replication and then harvested by centrifugation. After discarding the supernatant, cell pellets were flash frozen in liquid nitrogen. Genomic DNA was extracted using the DNeasy blood and tissue kit (Qiagen). qPCRs were performed using the Luna qPCR mix (NEB) in a Rotor-Gene Q Instrument (Qiagen). The *ori* region was amplified using primers 5'-GGCCATTGATCGTGCATCTC-3' and 5'-AGGTTCTGACAGGAAGGATA AGC-3' and the *ter* region using primers 5'-TTTGCATGAACTGGGC AATA-3' and 5'-TCCGAACATGTCCAATGAGA-3'. Using the Rotor-Gene Software version 2.3.5 (Qiagen) a relative quantification analysis (ΔΔC$_T$) was performed with the crossing points (C$_T$), averaged from three technical replicates, and PCR efficiency to determine *ori/ter* ratios. The *ori/ter* ratios were normalized to those of gDNA from *B. subtilis* spores which only have one chromosome and thus an *ori/ter* ratio of one. Experiments were independently performed twice.

### *B. subtilis* strains

*B. subtilis* strains are listed in Supplementary Data 1 and were propagated at 37 °C in Luria-Bertani (LB) medium unless stated otherwise in method details. Transformation of competent *B. subtilis* cells was performed using an optimized two-step starvation procedure[53,54]. Briefly, recipient strains were grown overnight at 37 °C in transformation medium (Spizizen salts supplemented with 1 μg/ml Fe-NH$_4$-citrate, 6 mM MgSO$_4$, 0.5% w/v glucose, 0.02 mg/ml tryptophan and 0.02% w/v casein hydrolysate) supplemented with 1% xylose where required. Overnight cultures were diluted 1:17 into fresh transformation medium supplemented with 1% xylose where required and grown at 37 °C for 3 h with continual shaking. An equal volume of prewarmed starvation medium (Spizizen salts supplemented with 6 mM MgSO$_4$ and 0.5% w/v glucose) was added and the culture was incubated at 37 °C for 2 h with continual shaking. DNA was added to 350 μl cells and the mixture was incubated at 37 °C for 1 h with continual shaking. 20–200 μl of each transformation was plated onto selective media supplemented with 1% xylose where required and incubated at 37 °C for 24-48 h. The genotype of all *dnaA* mutants was confirmed by DNA sequencing. Descriptions, where necessary, are provided below.

DnaA alanine substitution strains were generated by a blue/white screening assay using CW199 [*trpC2 dnaA::erm(dnaA$^{I/II}$ P$_{VEG}$-bgaB terminators incC ΔdnaN) amyE::spc(xylR P$_{XYL}$-dnaA-dnaN)*] as parental

strain and mutant plasmids obtained from pCW23 after Quickchange mutagenesis and sequencing as recombinant DNA. X-gal 0.016% w/v was added to transformation plates for detection of β-galactosidase activity and selection of chloramphenicol resistant white colonies that integrated mutant DNA by double-recombination. Eight individual white colonies per mutant were then restreaked onto a medium either with or without xylose to identify alleles of interest.

CW380 [*trpC2 amyE::(spc xylR $P_{XYL}$-dnaA dnaN tet)*] was constructed by transformation of pCW271 into CW59 [*trpC2 amyE::(spc xylR $P_{XYL}$-dnaA dnaN)*] and confirmed by sequencing.

FDS76 [*trpC2 aprE::kan(lacI $P_{spac}$-repN/oriN) amyE::spc($P_{xyl}$-dnaA dnaN) rpmH tetL incAB*] was generated by transforming sCW54 with a DNA fragment of the origin region containing a tetracycline resistance cassette integrated upstream of *incAB*. The fragment was generated by oTR002/oTR1219 PCR-amplifying a four-way Gibson Assembly of the following overlapping fragments: i) a plasmid backbone amplified from pCW270 with oFDS005/oFDS006, ii) a region upstream of *incAB* amplified from *B. subtilis* 168CA gDNA with oFDS048/oFDS318, iii) a tetracycline resistance cassette amplified from pCW270 with oFDS300/oFDS232, and iv) a region stretching from *incAB* to the 5′ end of *dnaN* amplified from *B. subtilis* 168CA gDNA with oFDS308/oFDS061. Transformants were confirmed by sequencing.

FDS257 [*trpC2 dnaA::erm(dnaA^{I/II} $P_{VEG}$-bgaB terminators incC ΔdnaN) amyE::(spc xylR $P_{XYL}$-dnaA dnaN tet)*] was constructed by transforming CW199 genomic DNA into CW380. Used as the parental strain to analyse expression of DnaA alanine variants by immunoblotting.

FDS404 [*trpC2 aprE::kan(lacI $P_{spac}$-repN/oriN) rpmH tetL incAB dnaA ΔincC dnaN cat*] was generated by transforming HM1108 with a DNA fragment of a tetracycline and chloramphenicol resistance cassette flanked origin region lacking *incC*. The fragment was generated by oFDS135/oFDS317 PCR-amplifying a two-way Gibson Assembly of the following overlapping fragments: (i) a region spanning from the 5′-half of *spoIIIJ* to the 5′-half of *dnaA* with a tetracycline resistance cassette being inserted between *rpmH* and *incAB*, amplified from sFDS076 gDNA with oFDS135/oFDS575, and (ii) a region spanning from the 3′-half of *dnaA* to the 5′-end of *recF* with *incC* being deleted and a chloramphenicol resistance cassette inserted between *dnaN* and *yaaA*, amplified from HM1603 gDNA with oFDS576/oFDS144. Transformants were confirmed by sequencing.

FDS682 [*trpC2 aprE::kan(lacI $P_{spac}$-repN/oriN) rpmH tetL incAB dnaA incC dnaN cat*] was generated by transforming HM1108 with a DNA fragment of a tetracycline and chloramphenicol resistance cassette flanked origin. The fragment was generated by oFDS135/oFDS317 PCR-amplifying a two-way Gibson Assembly of the following overlapping fragments: i) a region spanning from the 5′-half of *spoIIIJ* to the 5′-half of *dnaA* with a tetracycline resistance cassette being inserted between *rpmH* and *incAB*, amplified from sFDS076 gDNA with oFDS135/oFDS575, and ii) a region spanning from the 3′-half of *dnaA* to the 5′-end of *recF* and a chloramphenicol resistance cassette inserted between *dnaN* and *yaaA*, amplified from sTR011 gDNA with oFDS576/oFDS144. Transformants were confirmed by sequencing.

FDS686/FDS688/FDS691/FDS1029/FDS1031/FDS1035/FDS1037 [*trpC2 aprE::kan(lacI $P_{spac}$-repN/oriN) rpmH tetL incAB dnaA incC(DnaA-trio$_{1-3}$ central A->C/G/T; DnaA-trio$_3$ first A->C/G/T; DnaA-trio$_{1-6}$ all first positions A) dnaN cat*] were generated by transforming HM1108 with DNA fragments of a tetracycline and chloramphenicol resistance cassette flanked origin in which the trio sequence has been mutated. The respective DNA fragments were generated by oFDS136/oFDS317 PCR-amplifying two-way Gibson Assemblies of the following overlapping fragments: i) a tetracycline resistance-marked region spanning from the 5′-half of *spoIIIJ* to *incC* introducing the trio mutations in the fragment overlap region amplified from sFDS682 gDNA with either oFDS135/oFDS1285 (for FDS686), oFDS135/oFDS1289 (for FDS688), oFDS135/oFDS1293 (for FDS691), oFDS135/oFDS1652 (for FDS1029),

oFDS135/oFDS1654 (for FDS1031), oFDS135/oFDS1656 (for FDS1035) or oFDS135/oFDS1658 (for FDS1037), and ii) a chloramphenicol resistance-marked region spanning from *incC* to the 5′-end of *recF*, introducing the mutations in the fragment overlap region, amplified from sTR011 gDNA with either oFDS1286/oFDS144 (for FDS686), oFDS1290/oFDS144 (for FDS688), oFDS1294/oFDS144 (for FDS691), oFDS1653/oFDS144 (for FDS1029), oFDS1655/oFDS144 (FDS1031), oFDS1657/oFDS144 (for FDS1035) or oFDS1659/oFDS144 (for FDS1037). Transformants were confirmed by sequencing.

TR011 [*trpC2 dnaN cat*] was generated by transforming *B. subtilis* 168CA with pHM327.

### *E. coli* strains and plasmids

*E. coli* transformations were performed in DH5α via heat shock following the Hanahan method and propagated in LB with appropriate antibiotics at 37 °C. Plasmids are listed in Supplementary Data 1 (sequences are available upon request). Descriptions, where necessary, are provided below.

pCW23 was constructed by Gibson assembly using a low-copy number plasmid backbone from pBR322 amplified using oCW83/oCW460, the *dnaA-incC-dnaN′* region from *B. subtilis* genomic DNA amplified with oCW86/oCW80 and the *'dnaN cat yaaA-recF'* region from pHM327 amplified with oCW79/oCW82.

pCW173 was constructed by Gibson assembly using pCW23 as template to amplify the plasmid backbone with oCW491/458, the *incC* region from *B. subtilis* 168CA gDNA with oCW481/490, $P_{VEG}$-bgaB from CW197 gDNA with oCW459/464, terminators with oCW471/472 and the erythromycin resistance cassette from pAPNC213-*erm* with oCW461/474.

pCW271 was constructed by Gibson assembly using pCW270 as template for *amyE* integration with a tetracycline resistance cassette amplified with oCW680/678 and the *'dnaA dnaN'* fragment amplified from *B. subtilis* 168CA gDNA with oCW677/oCW683.

DnaA alanine-scan mutant plasmids were generated by Quickchange mutagenesis using oligonucleotides listed in Supplementary Data 1. Cloning protocols were adapted to a 96-well plate format for PCR amplification of mutant plasmids, heat-shock and transformation recovery. All plasmids were sequenced.

### Oligonucleotides

All oligonucleotides were purchased from Eurogentec. Oligonucleotides used for plasmid construction and those generated by the Quickchange program are listed in Supplementary Data 1. Quickchange mutagenesis was used for the construction of the DnaA alanine scan library. Each point mutant was assembled by PCR using mutagenic primers carrying a single alanine substitution[55]. We generated all mutant primer pairs via an in-house Quickchange program with identical parameters as used previously[56]. The code was written in Java and is available online (https://zenodo.org/record/5541537#.Yz1lQHbMJHY).

### Reporting summary

Further information on research design is available in the Nature Portfolio Reporting Summary linked to this article.

## Data availability

The previously published X-ray data used in this study are available in the PDB database under accession code 3R8F. The cryo-electron microscopy data generated in this study have been deposited in the Electron Microscopy Data Bank under accession codes EMD-16230 (domain III lattice), EMD-16231 (dsDNA region), EMD-16256 (domain IV lattice), and EMD-16229 (overall map). The X-ray data generated in this study is available in the PDB database under accession code 8BV3. The structural model of the BUS complex generated in this study has been deposited in the PDB database under accession code 8BTG. Source data are provided with this paper.

## Code availability

Code used in the generation of data from single molecule fluorescence microscopy experiments is available at https://github.com/GMerces/DisappearingSpots/tree/main under a GPL-3.0 license (https://doi.org/10.5281/zenodo.8363454). The code is designed for use with FIJI using the ImageJ Macro language. Code used to generate mutagenic primer pairs for the DnaA alanine scan is available at https://zenodo.org/record/5541537#.Yz1lQHbMJHY under a Creative Commons Attribution 4.0 International license (https://doi.org/10.5281/zenodo.5541537). Use of these codes is permitted without limitation, as is modification of the codes as desired.

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

## Acknowledgements

The authors thank Tiago Costa, Jeff Errington, Waldemar Vollmer, and Gabriel Waksman for a critical review of the manuscript. We thank Frances Davison and Arnaud Baslé for their technical assistance. We acknowledge Diamond Light Source for access and support of the cryo-EM facilities at the UK National Electron Bio-imaging Centre (eBIC, proposal BI25832), funded by the Wellcome Trust, MRC, and BBSRC. The Jeol 2100 plus TEM used for cryo-EM grid screening at QMUL was funded by an UKRI Altert16 grant (BB/R000514/1). We acknowledge the Diamond Light Source (Didcot, UK) for beam time (BAG Proposal mx24948), and thank the staff of Beamline I24 for support. Research support was provided to HM by a Wellcome Trust Senior Research Fellowship (204985/Z/16/Z), to AI by an Academy of Medical Sciences Springboard Award (SBF007\100161), and to both HM and AI by a Wellcome Trust Discovery Award (225811/Z/22/Z). HY and DRB were supported by the Francis Crick Institute, which receives core funding from Cancer Research UK (FC001221), the UK Medical Research Council (FC001221), and the Wellcome Trust (FC001221). FDS was supported by a post-doctoral research fellowship from the Deutsche Forschungsgemeinschaft (SCHR 1684/1-1).

## Author contributions

S. Pelliciari, D.R.B., C.W., F.D.S., H.Y., A.I., H.M. conceived the research plan. Pelliciari, S.B.-L., S.F., D.S., C.W., F.D.S., S. Pintar., T.T.R., Y.T., J.H., A.I. generated results presented in the manuscript. G.M. developed the image analysis pipeline and assisted with the image analysis. S. Pelliciari, C.W., F.D.S., S. Pintar, A.I., H.M. created figures. S. Pelliciari, A.I., H.M. wrote the manuscript. S. Pelliciari, H.Y., C.W., F.D.S., A.I., H.M. edited the manuscript.

## Competing interests

The authors declare no competing interests.
