## [Peer Review File · Nature Communications]

REVIEWER COMMENTS

Reviewer #1 (Remarks to the Author):

In this manuscript entitled "Opening a bacterial replication origin" Peliiciari et al., attempted to study the molecular basis of how DnaA interacts with a conserved trio sequence present in the bacterial chromosomal origin. Using the structure-function approach, authors showed that BsDnaAATP specific oligomer promotes origin opening at specific DNA-trios region using a base-flipping mechanism. Authors suggested that A17 and G18 in the trio's sequence are flipped into the nucleotide-binding pocket, which is created by the two adjacent DnaA protomers. A detailed study of nucleoprotein complexes involved in DNA duplex opening is required to understand the fundamental cellular processes of Chromosomal replication and how it is controlled so tightly during the cell cycle. The present study in parts is interesting and provides important information to understand duplex opening by DnaA protein.

Major comments:

The authors can structure the introduction better. The introduction delivers minimal information for a reader outside the Bacterial DNA replication field. Considering the Journal has a broad readership, the authors should include better details about the origin opening, the trio's sequences, and DnaA box sequences, including DnaA boxes 6 & 7 (the basis of the manuscript). The start of the introduction section is misleading to the reader, with the initial paragraph providing the impression that the study may relate to Helicases and the Helicase loading aspect of the initiation complex formation. Since the primary focus of the manuscript is DNA Duplex opening (a pre-helicase recruitment step), the initial paragraph of the introduction needed to be reduced and placed later in the section, wherever it is appropriate. Additionally, the last part of the introduction should be more elaborative and include the unknowns that the authors wanted to address in the present study and the approach authors have chosen to address those unknowns. The title of the manuscript needs to be more specific. The opening of the DNA duplex includes several steps, in addition to the involvement of the studied trio's sequences in promoting the unwinding of the chromosomal origin by the DnaA protein.

The authors did not advocate for any concrete reason to exclude the possibility that DNA strand separation may not follow the trans mechanism. Later in the manuscript, the authors did not touch the cis/trans-binding aspect, likely losing the importance of the information obtained in Fig.1 and S1-S4. The relevance of this study is to understand how base flipping promotes origin unwinding, and at least the present study showed no correlation between the cis/trans mechanism and base flipping.

The experiments designed and performed to generate data in Fig1 and S2-S4 were only a slight modification of the previously published data (Fig. 2: NAR 2021 by Simone Pellicciari). These experiments suggest that DnaAATP cause the unwinding of DNA duplex resulting in the dropping of fluorescent-labeled DNA probe, which is complimentary to the trio's region. How these experiments indicated that unwinding occurs via the cis/trans mechanism is complicated to understand. If it has something to do with chosen parameters such as the distance between DNA scaffolds. However, confirming this requires supporting evidence. For example, the hybrid DNA fragment carrying dsDNA regions (with DnaA boxes, 6&7) and ssDNA region (with DNATrios), labeled with two labels (either radioactive or fluorescent) following mixing with a complementary strand (ATTO-695 labeled) to form a full-length double strand. The addition of DnaAATP unwinds DNA duplex, thus dropping ATTO-695 fluorescence. In addition, in the case of the cis mechanism, the remaining complex should ideally contain single-labeled hybrid DNA. Otherwise, a mix of labeled DNA hybrid should be present, possibly confirming that unwinding may occur via the trans mechanism.

The authors only discussed the interaction between DnaA boxes (6&7) and DnaA1-2 without commenting on the interactions of dsDNA with proximal protomers. Does that mean only the frontal protomer of the DnaA helix interacts with DnaA boxes (6&7) present within the dsDNA? Explain.

So much is in the literature related to the involvement of residues residing in the AAA+ domain and the DNA-binding domain IV. Moreover, the mechanisms that explain how changing crucial residues in AAA+ and DNA binding domains cause initiation defects in the DnaA protein were also reported. In random Ala screening, it is no wonder that the authors found 37 amino acid residues where mutation causes growth defects. Only modeling studies combining the X-ray crystal structure of the AAA+ domain with cryoEM structures could identify residues Lys222, Glu223, Arg202, Glu228, Glu183, Ile190, and Ile193. A more reasonable approach to test is the functions that will be defective when these residues are changed and how those functions are relevant to the DnaA-BUS interaction model. The authors need to add supportive experiments to explain what initiator functions (ATP-binding, DnaA-DnaA oligomerization at BUS, or opening of DNA duplex-via trans/cis mechanism) in DnaA protein are defective and how these functions correlate with the DnaA-BUS interaction model. Otherwise, the results presented in these experiments are of limited interest.

Out of six trio sequences, five repeats (one, two, four, five, and six) contain GA, which flips inside. It is only the third repeat, which has AA. As per the author's work, the two bases that flipped between adjacent protomers can be A-G (inserted between DnaA1-2) or A-A (inserted between DnaA2-3). The authors should examine if replacing one A in the third repeat by G is tolerated in the origin-unwinding model. It will highlight the specificity of adenine and guanine positions in each repeat in the base-flipping model.

If there were no effect of substitution of Glu183 and Glu223 by Ala on growth, how are the interactions of these amino acids with dsDNA (Glu223 with G19) or ssDNA (Glu183 with A17) important for base flipping and DNA duplex opening? Explain.

Minor comments:

No need for repetitive figures. The repetitive part of Fig. 1A-B, S2 & S3A should be merged to make one figure (present it in the main section or supplementary, wherever found appropriate).

S3B and S3C should be part of Fig. 1.

Again, S1B and 3E are the same Figures.

The font size of Fig. S7E is so small that it is hard to read the labels.

Fig. S7A: numbers under color shade are missing.

Reviewer #2 (Remarks to the Author):

This is a follow-up study by the Murry lab. They previously identified a basal unwinding system (BUS) including a repeating trinucleotide motif termed DnaA-trios that initiator DnaA recognizes at the *B. subtilis* replication origin (Richardson et al. Nature 2016). In this paper, they have combined in vitro single-molecular and in vivo mutagenic studies with structural visualization to reveal the mechanism of origin unwinding by the *B. subtilis* initiator DnaA. Their work largely confirms the previous conclusion by the Berger lab that DnaA oligomerizes into a spiral to stretch on ssDNA and that each alpha/beta-folded ATPase domain (Domain III) contacts three nucleotide bases during initial origin melting (Duderstadt et al. Nature 2011). However, the current work used a true origin DNA sequence of *B. subtilis*, contrary to the artificial poly-A sequence used by the Berger lab, and revealed some interesting details. Therefore, this work may be of sufficient interest for Nature Communications, after several issues raised below are addressed.

Major issue:

1. It should be specified in title or at least in the abstract that this work pertains to *B. subtilis*. This is important because bacterial origins are diverse, and it is unclear if the most studied model bacterium *E. coli* uses the exactly the same mechanism.

2. Base flipping. The authors may have made too big a deal about the two of the trio bases being “flipped”. The use of this wording is questionable and potentially misleading, because “base flipping” usually refers to a base being flipped out of a dsDNA. However, in the current structure, the so-called “flipped bases” are in the ssDNA region and the bases can be stabilized at any positions. In other words, the exact position of stabilized base may not be critical. Consistent with this

observation, the two hydrophobic residues that sandwich the “flipped bases” Ile190 and Ile193 (Figure S14) are not essential to cell growth (Fig. S11C). The authors are suggested to avoid the term “base flip” and tune down the significance of the exact base positions.

3. The authors used a tailed DNA (BUS sequence with a single-stranded DnaA-trios). So, they did not capture a true unwinding state (as their substrate is already “unwound” before the addition of DnaA). The authors should at least try to start with dsDNA and to capture the unwinding product by cryo-EM, which would be more meaningful.

4. Fig. S1A should be moved to be part of main text Fig. 1 as it is critical to understanding the structural and smTIRF experiments, as well as to providing context of previous X-ray structural analysis with poly-A sequence.

5. The dsDNA binding region is at 5 Å resolution. Explain how the DNA sequence was assigned in this region.

Minor issue:

1. In Figs. S11 and S12, the wild-type control is strain CW233, while the parent strain used for those mutations is CW199, please explain the rationale for using different strains.

2. Fig. S11C appears to contain multiple splicing. If true, either leave space in between splices or state so in legend.

3. Fig. S7A, some words and numbers at the bottom of the panel are blocked and invisible.

4. Ile346 should be highlighted in Fig. 2E (P8, line 175).

5. The method section needs more details. For example, the details for reconstituting the BUS complex. What was the used molar ratio of protein and DNA? How was the cross-linking experiment performed? What were the buffer, reagent, and protein concentration and ratio?

Typos

Line 124, higher should be “lower”

Line 135, 5.9 should be “5.4”

Line 160, grove should be “groove”

Line 197, missing the label of α -15 in Fig. 3A

Line 223, Figs should be “Fig” and Line 228 vice versa

Line 246, Fig. 3C should be Fig. 3B.

Line 345, inc. should be “Inc.” (also in Line 370), and NaCO₃ should Na₂CO₃

Line 405, QIAgen use either the initial uppercase or entire

Line 420, missing a space between number and the symbol of unit, also in Line 425, 426, 431, 434, 435, 453, 476, 480, 571, 580, 598, 615, 618, 623, 625, 627, 629, 705

Line 448, Tev - all capital

Line 490, CryoSparc should be cryoSPARC, also in Line 499, 510

Line 509, higher should be lower

Line 499-514, missing comma in particle numbers

Line 515, 6.1 Å should be 5.4 Å

Figures and legends:

Fig. 2E, the green label should be "DnaA2"

Fig. S5C, labels of protein markers are missed

Fig. S11B, use consistent presentation of alanine mutations, either add or remove all "A" after the residue numbers

Table S3, add comma to final particle numbers

Table S4, Ramachandran plot parameters are missing

Reviewer #3 (Remarks to the Author):

The present work by Pelliciani et al. combines multiple techniques, including single molecule and cryo-EM, to establish a molecular mechanism for the unwinding of a bacterial replication origin. Using a short overhang substrate that mimics an unwound state of origin DNA, the authors imaged a DnaA oligomer bound to it using cryo-electron microscopy. The resulting structure explains the specificity of DnaA protein for origin DNA – in particular 'DnaA trio' sequences – as well as highlighting a new DNA engagement mode for the duplex DNA binding region (domain IV) of the protein.

The imaged complex significantly advances our understanding of how DnaA destabilizes and unwinds origin DNA in preparation for replicative helicase loading, in particular showing that DnaA uses a base flipping mechanism to promote local melting. The structure, together with single molecule imaging, is consistent with a proposed 'templated-extension' model for origin firing in which one type of DnaA oligomer bind dsDNA and nucleates filament assembly by a second set of DnaA protomers that melt the origin and bind ssDNA; these findings similarly refute a 'loop-back' model which has suggested that dsDNA-bound DnaA oligomers can also bind ssDNA following

melting. Given its impact, the work should be published in Nature Communications. Only a few minor revisions are suggested:

- It seems surprising that domain IV of subunit 1 binds to the dsDNA A box that is closer to the ssDNA junction than subunit, which binds further away. What does this imply for the binding of neighboring (upstream) DnaA boxes in terms of how their domain III elements would interact with other AAA+ folds?
- Given the significance of this DnaA-trios from this work, how do authors reconcile this with previous work in *E. coli* (Oyman et. Al, FASEB Journal, 2006) in which authors show that DnaA trios can be mutated without any effect?
- The Methods section should include the purification details for DnaA104-446.
- The Methods section should describe complex formation and crosslinking conditions for generating the DnaA-DNA complex in detail.
- In Figure S5B, most of the DNA seems to be unbound. Please comment.
- Fig. S2 is called out before Fig. S1B - this is a bit confusing.
- Fig. S2. The diagram for DNA unwinding in cis is a little confusing. It would be helpful to flip the first two icons for DnaA upside down so that they can nucleate the ssDNA-bound filament. As it stands, the schematic gives the impression that the two sets of complexes are fully distinct and do not engage one another, which the structure shows isn't the case.
- Figs. S5-S6 - a plot of particle distribution is needed to assess preferred orientation bias.
- Line 136, Fig. S8. The resolution of the X-ray model should be stated in the text and a representative view of electron density included in the supplemental figure.
- It would be helpful to have a movie that runs through the various organizational aspects of the EM model.
- Line 146. Perhaps "confirms" would be a better word choice than "resolves"?
- Line 197. Please label alpha-15 on the figure.
- Fig. S1B. Please include *E. coli*, *A. aeolicus*, *Mtb*, and *T. maritima* sequences, as these are commonly studied species in the literature.
- Figs. S13 and 14. Please show protein side chains colored by atom as they are for the DNA.

REVIEWER COMMENTS: NCOMMS-23-06337-T

We thank the Reviewers and the Editor for their time and effort. We have tried to address all the comments, concerns, and constructive criticism raised. Please note that references to Figures correspond to the current resubmitted version.

Reviewer #1:

Major comments:

The authors can structure the introduction better. The introduction delivers minimal information for a reader outside the Bacterial DNA replication field. Considering the Journal has a broad readership, the authors should include better details about the origin opening, the trio's sequences, and DnaA box sequences, including DnaA boxes 6 & 7 (the basis of the manuscript). The start of the introduction section is misleading to the reader, with the initial paragraph providing the impression that the study may relate to Helicases and the Helicase loading aspect of the initiation complex formation. Since the primary focus of the manuscript is DNA Duplex opening (a pre-helicase recruitment step), the initial paragraph of the introduction needed to be reduced and placed later in the section, wherever it is appropriate. Additionally, the last part of the introduction should be more elaborative and include the unknowns that the authors wanted to address in the present study and the approach authors have chosen to address those unknowns. The title of the manuscript needs to be more specific. The opening of the DNA duplex includes several steps, in addition to the involvement of the studied trio's sequences in promoting the unwinding of the chromosomal origin by the DnaA protein.

We have modified the Introduction by adding more information and focusing on DNA opening. We have also elaborated on the critical questions to be addressed and the methods that were used. As suggested, we have modified the Title to be more specific.

The authors did not advocate for any concrete reason to exclude the possibility that DNA strand separation may not follow the trans mechanism. Later in the manuscript, the authors did not touch the cis/trans-binding aspect, likely losing the importance of the information obtained in Fig.1 and S1-S4. The relevance of this study is to understand how base flipping promotes origin unwinding, and at least the present study showed no correlation between the cis/trans mechanism and base flipping.

We do not wish to exclude the *in trans* model at all, especially for bacterial species with divergent replication origins. Importantly however, observing that BUS strand separation activity was possible *in cis* justified and motivated our efforts to pursue single-particle structural analysis using the simplified DNA scaffolds. To link this point to the broader study, we have integrated the *cis/trans* topic into the discussion section entitled "Model for BUS base capture and origin opening".

As noted by Review#2, this work does not address how DnaA base flipping promotes origin unwinding. Rather it provides an explanation for how DnaA-trios are specifically recognized by a DnaA oligomer anchored at DnaA-boxes (i.e. the BUS nucleoprotein complex). The observation that the nucleobases are captured within a binding pocket strongly suggests a role in unwinding, but this will require further experimental support, which is beyond the scope of this study.

The experiments designed and performed to generate data in Fig1 and S2-S4 were only a slight modification of the previously published data (Fig. 2: NAR 2021 by Simone Pellicciari). These experiments suggest that DnaAATP cause the unwinding of DNA duplex resulting in the dropping of fluorescent-labeled DNA probe, which is complementary to the trio's region. How these experiments indicated that unwinding occurs via the cis/trans mechanism is complicated to understand. If it has something to do with chosen parameters such as the distance between DNA scaffolds.

While the DnaA strand separation assay is similar to the previously published work, the detection method using single molecule imaging is novel and allows us to study/characterize these events on individual DNA template without the influence of other components, largely eliminating any unnatural avidity effect from adjacent complexes.

We have modified the text to explain the rationale of the smTIRF assays and how they support the *in cis* model. In our experimental set-up we use a flow cell to deliver substrate molecules onto the derivatized glass coverslip. First, the fluorescently labelled DNA scaffolds are immobilized using streptavidin:biotin capture. Second, all unbound DNA molecules are washed away using the flow cell, therefore the only DNA substrates available for strand separation are those that remain immobilized. Because we know the length of our individual DNA scaffolds and we can determine the minimal distances between fluorescently labelled

substrates, we conclude that strand separation must occur *in cis* and not between two DNA scaffolds. Supportively, the cryo-EM map provides a molecular mechanism for strand separation *in cis*.

However, confirming this requires supporting evidence. For example, the hybrid DNA fragment carrying dsDNA regions (with DnaA boxes, 6&7) and ssDNA region (with DNATrios), labeled with two labels (either radioactive or fluorescent) following mixing with a complementary strand (ATTO-695 labeled) to form a full-length double strand. The addition of DnaAATP unwinds DNA duplex, thus dropping ATTO-695 fluorescence. In addition, in the case of the *cis* mechanism, the remaining complex should ideally contain single-labeled hybrid DNA. Otherwise, a mix of labeled DNA hybrid should be present, possibly confirming that unwinding may occur via the *trans* mechanism.

A similar experiment was performed and is shown in Figure S3. Here two of the oligonucleotides in the DNA scaffold were fluorescently labelled, and consistent with the *in cis* model we observed specific loss of the oligo complementary to the DnaA-trios (ATTO 647), while the oligo containing the DnaA-trios remains relatively stable (ATTO 565). Note that the signal from the oligo labelled with ATTO 565 indicates that the annealed biotin-labelled oligo must also be present, otherwise the scaffold would not have been immobilized.

The authors only discussed the interaction between DnaA boxes (6&7) and DnaA1-2 without commenting on the interactions of dsDNA with proximal protomers. Does that mean only the frontal protomer of the DnaA helix interacts with DnaA boxes (6&7) present within the dsDNA? Explain.

The cryo-EM map does not provide any evidence to support an interaction of DnaA₄₋₇ with dsDNA. Line 185 now includes the statement: "No interaction was detected between protomers DnaA₄₋₇ and dsDNA." We have also added a paragraph to the discussion addressing dsDNA binding and additional DnaA-boxes.

So much is in the literature related to the involvement of residues residing in the AAA+ domain and the DNA-binding domain IV. Moreover, the mechanisms that explain how changing crucial residues in AAA+ and DNA binding domains cause initiation defects in the DnaA protein were also reported. In random Ala screening, it is no wonder that the authors found 37 amino acid residues where mutation causes growth defects. Only modeling studies combining the X-ray crystal structure of the AAA+ domain with cryoEM structures could identify residues Lys222, Glu223, Arg202, Glu228, Glu183, Ile190, and Ile193. A more reasonable approach to test is the functions that will be defective when these residues are changed and how those functions are relevant to the DnaA-BUS interaction model. The authors need to add supportive experiments to explain what initiator functions (ATP-binding, DnaA-DnaA oligomerization at BUS, or opening of DNA duplex-via *trans/cis* mechanism) in DnaA protein are defective and how these functions correlate with the DnaA-BUS interaction model. Otherwise, the results presented in these experiments are of limited interest.

During the course of this study we considered these experiments, especially for the essential residues implicated in base interactions (N187, R202, E228). However, the ISM, in addition to forming the dinucleotide binding pocket, also forms an important protein:protein interface between adjacent DnaA protomers. Key residues implicated in base interactions from the BUS structure presented here (N187, R202) had previously been changed to alanine in the *Aquifex* DnaA homolog (Duderstadt 2010. doi:10.1074/jbc.M110.147975), and it was found that these DnaA variants (*Aquifex* Q153A, R168A) were defective in oligomer assembly (see Figure 4B). If DnaA cannot form an oligomer, then it cannot assemble the dinucleotide binding pocket, thus preventing the ability to capture nucleobases. It was for these reasons that we pursued a chemical genetic approach to modify the nucleobases, rather than the protein.

We envision that it may be possible to mutate these ISM residues and retain protein activities (e.g. compensatory changes involving multiple residues at the protein:protein interface). However, this will likely involve significant experimental work to identify useful DnaA variants, which is beyond the scope of this manuscript. We would also note that the only assay currently available to detect DnaA base capture is cryo-EM. Testing DnaA mutants for base flipping defects using this structural approach is not feasible in a timely manner. Future work will be directed towards this aim, as well as towards developing biophysical assays to detect base flipping by DnaA.

Out of six trio sequences, five repeats (one, two, four, five, and six) contain GA, which flips inside. It is only the third repeat, which has AA. As per the author's work, the two bases that flipped between adjacent protomers can be A-G (inserted between DnaA1-2) or A-A (inserted between DnaA2-3). The authors should examine if replacing one A in the third repeat by G is tolerated in the origin-unwinding model. It will highlight the specificity of adenine and guanine positions in each repeat in the base-flipping model.

This is an interesting hypothesis which we tested by creating two derivative strains, in which all DnaA-trios were mutated to encode either GA or AA. While the GA set was well tolerated, the AA set resulted in a

DNA replication initiation defect. Therefore, we conclude that while AA can be present at certain positions, GA is the preferred dinucleotide pair for function (presumably base capture). We have added these new results to the manuscript (see Fig. S14).

If there were no effect of substitution of Glu183 and Glu223 by Ala on growth, how are the interactions of these amino acids with dsDNA (Glu223 with G19) or ssDNA (Glu183 with A17) important for base flipping and DNA duplex opening? Explain.

The DnaA^{E183A} variant does produce a mild growth defect, but this interaction is not essential for viability under the conditions tested. It appears that DnaA^{N187} plays the dominant role in the proposed nucleobase capture mechanism. While E183 does likely contribute to base capture, we have modified the text throughout to focus on the essential residue N187.

DnaA^{E223} contacts the phosphodiester bond between G19 and G20 using the main chain oxygen rather than the side chain. This likely explains why the DnaA^{E223A} variant does not produce a detectable phenotype.

Minor comments:

No need for repetitive figures. The repetitive part of Fig. 1A-B, S2 & S3A should be merged to make one figure (present it in the main section or supplementary, wherever found appropriate).

Done

S3B and S3C should be part of Fig. 1.

These have been moved.

Again, S1B and 3E are the same Figures.

These ISM alignments highlight a different set of species, often related to previous studies or collections. Figure 3E displays the ISM from DnaA homologs that have been shown to constitute a functional BUS for strand separation *in vitro*. Figure S1A displays the ISM from DnaA homologs from ESKAPE pathogens. And we have now added the ISM from commonly used model organisms to Figure S1 (as requested by Reviewer#3). Beyond showing *B. subtilis* as a reference sequence, the only repeated sequence between these sets is *Staphylococcus aureus*.

The font size of Fig. S7E is so small that it is hard to read the labels.

Done

Fig. S7A: numbers under color shade are missing.

Done

Reviewer #2:

Major issue:

1. It should be specified in title or at least in the abstract that this work pertains to *B. subtilis*. This is important because bacterial origins are diverse, and it is unclear if the most studied model bacterium *E. coli* uses the exactly the same mechanism.

To be more precise, the title has been changed to “The bacterial replication origin BUS promotes nucleobase capture” and the abstract states that the experimental work utilized *Bacillus subtilis*.

2. Base flipping. The authors may have made too big a deal about the two of the trio bases being “flipped”. The use of this wording is questionable and potentially misleading, because “base flipping” usually refers to a base being flipped out of a dsDNA. However, in the current structure, the so-called “flipped bases” are in the ssDNA region and the bases can be stabilized at any positions. In other words, the exact position of stabilized base may not be critical. Consistent with this observation, the two hydrophobic residues that sandwich the “flipped bases” Ile190 and Ile193 (Figure S14) are not essential to cell growth (Fig. S11C). The authors are suggested to avoid the term “base flip” and tune down the significance of the exact base positions.

We agree that base flipping out of dsDNA has not been observed, therefore we have tempered our interpretation of dsDNA opening.

However, please note that both DnaA^{I190A} and DnaA^{I193A} variants are lethal *in vivo* (Figures S11B and S12A), therefore these small hydrophobic residues are essential for DnaA function. Taken together with the chemical genetic evidence supporting the specificity of the captured nucleobases, we believe that the observed position of the proposed bases is significant.

3. The authors used a tailed DNA (BUS sequence with a single-stranded DnaA-trios). So, they did not capture a true unwinding state (as their substrate is already “unwound” before the addition of DnaA). The authors should at least try to start with dsDNA and to capture the unwinding product by cryo-EM, which would be more meaningful.

Indeed, experiments to isolate and image such dsDNA nucleoprotein complex intermediates are underway. However, we believe that this is beyond the scope of the current study, which is aimed at elucidating the specificity of DnaA for the BUS sequence elements within *oriC*.

4. Fig. S1A should be moved to be part of main text Fig. 1 as it is critical to understanding the structural and smTIRF experiments, as well as to providing context of previous X-ray structural analysis with poly-A sequence.

Done

5. The dsDNA binding region is at 5 Å resolution. Explain how the DNA sequence was assigned in this region.

Though the resolution is not good enough to see individual bases within the dsDNA region, we can see distinct backbone phosphate density. This allowed us to place a B-form DNA with the right sequence within the density as a rigid unit. The correct assignment of the sequence to the model was checked from both ends, i.e. the ssDNA 5' end where the resolution is much higher and from the dsDNA end. Below are some representative figures that shows the dsDNA B-DNA placed within the density. Note at the dsDNA end, the 5' phosphate can be seen for one strand, but not the other, which further aided us to place the correct strand within its density.

Minor issue:

1. In Figs. S11 and S12, the wild-type control is strain CW233, while the parent strain used for those mutations is CW199, please explain the rationale for using different strains.

CW233 is a derivative of CW199 (the same as all alanine mutants). It was generated by transforming the wild-type plasmid using for Quickchange mutagenesis (pCW23) into CW199. This information has been added to the strain list in Table 1.

2. Fig. S11C appears to contain multiple splicing. If true, either leave space in between splices or state so in legend.

This is correct and is now stated in the figure legend.

3. Fig. S7A, some words and numbers at the bottom of the panel are blocked and invisible.

Done

4. Ile346 should be highlighted in Fig. 2E (P8, line 175).

Done

5. The method section needs more details. For example, the details for reconstituting the BUS complex. What was the used molar ratio of protein and DNA? How was the cross-linking experiment performed? What were the buffer, reagent, and protein concentration and ratio?

Done

Typos

Line 124, higher should be "lower"

Done

Line 135, 5.9 should be "5.4"

Done

Line 160, grove should be "groove"

Done

Line 197, missing the label of α -15 in Fig. 3A

Done

Line 223, Figs should be "Fig" and Line 228 vice versa

Done

Line 246, Fig. 3C should be Fig. 3B.

Done

Line 345, inc. should be "Inc." (also in Line 370), and NaCO₃ should Na₂CO₃

Done

Line 405, QIAgen use either the initial uppercase or entire

Done

Line 420, missing a space between number and the symbol of unit, also in Line 425, 426, 431, 434, 435, 453, 476, 480, 571, 580, 598, 615, 618, 623, 625, 627, 629, 705

Done

Line 448, Tev - all capital

Done

Line 490, CryoSparc should be cryoSPARC, also in Line 499, 510

Done

Line 509, higher should be lower

Done

Line 499-514, missing comma in particle numbers

Done

Line 515, 6.1 Å should be 5.4 Å

Done

Figures and legends:

Fig. 2E, the green label should be "DnaA2"

Done

Fig. S5C, labels of protein markers are missed

Added

Fig. S11B, use consistent presentation of alanine mutations, either add or remove all "A" after the residue numbers

Added to all.

Table S3, add comma to final particle numbers

Done

Table S4, Ramachandran plot parameters are missing

The Ramachandran plot data has been added to the Methods.

Reviewer #3:

- It seems surprising that domain IV of subunit 1 binds to the dsDNA A box that is closer to the ssDNA junction than subunit, which binds further away. What does this imply for the binding of neighboring (upstream) DnaA boxes in terms of how their domain III elements would interact with other AAA+ folds?

We have added a paragraph to the discussion regarding upstream DnaA-boxes. Because there are many of these sites within bacterial origins, they may facilitate interactions through more complex nucleoprotein structures.

- Given the significance of this DnaA-trios from this work, how do authors reconcile this with previous work in *E. coli* (Oyman et. Al, FASEB Journal, 2006) in which authors show that DnaA trios can be mutated without any effect?

We believe this reference relates to an abstract from the Experimental Biology 2019 Meeting: Oyman, Rao, Grimwade, Leonard "Mutational Analysis of DnaA-trio Motifs in *E. coli* OriC Reveals Multiple Modes for Bacterial Replication Origin Activation" doi.org/10.1096/fasebj.2019.33.1_supplement.619.8. There is no full text article associated with this abstract published in *The FASEB Journal*, therefore we cannot comment on its relevance.

We do appreciate this point regarding *E. coli* and, and as we have published previously, there is evidence that the *E. coli* chromosome origin has diverged from the ancestral BUS (Richardson 2019 doi.org/10.15252/embj.2019101649).

- The Methods section should include the purification details for DnaA104-446.

Done

- The Methods section should describe complex formation and crosslinking conditions for generating the DnaA-DNA complex in detail.

Done

- In Figure S5B, most of the DNA seems to be unbound. Please comment.

Apologies, while this peak likely does contain DNA and ATP, the majority of the absorbance signal turns out to be from the glutaraldehyde used for crosslinking (see figure below using reaction buffer with specific supplements: ATP is blue, glutaraldehyde is orange, DNA is green). We have relabelled this peak as "x-linker".

- Fig. S2 is called out before Fig. S1B - this is a bit confusing.

Figures have been reordered.

- Fig. S2. The diagram for DNA unwinding in cis is a little confusing. It would be helpful to flip the first two icons for DnaA upside down so that they can nucleate the ssDNA-bound filament. As it stands, the schematic gives the impression that the two sets of complexes are fully distinct and do not engage one another, which the structure shows isn't the case.

Done

- Figs. S5-S6 - a plot of particle distribution is needed to assess preferred orientation bias.

This has been added to Figure S5.

- Line 136, Fig. S8. The resolution of the X-ray model should be stated in the text and a representative view of electron density included in the supplemental figure.

Both have now been added to the supplemental figure.

- It would be helpful to have a movie that runs through the various organizational aspects of the EM model. We have added three movies to the manuscript, one progressing through the various organizational aspects of the cryo-EM model, a second highlighting the interactions with dsDNA and the third focusing on a single trio showing the captured 2 bases by the ISM.

- Line 146. Perhaps "confirms" would be a better word choice than "resolves"?

Line 165: We have modified the text to "This orientation of the DnaA oligomer on the DnaA-trios is compatible with the proposition that DnaA interacts directly with the AAA+ motif of a loader protein to guide helicase deposition."

- Line 197. Please label alpha-15 on the figure.

Done

- Fig. S1B. Please include *E. coli*, *A. aeolicus*, *Mtb*, and *T. maritima* sequences, as these are commonly studied species in the literature.

Done

- Figs. S13 and 14. Please show protein side chains colored by atom as they are for the DNA.

Done

REVIEWERS' COMMENTS

Reviewer #1 (Remarks to the Author):

Dear Editor,

The revised manuscript submitted by the authors has addressed the issues raised in the initial submission. I recommend acceptance of the revised version for publication in the Nature communication journal.

Thank you very much.

Warm regards

Reviewer #2 (Remarks to the Author):

The authors have addressed the issues we raised.

Reviewer #3 (Remarks to the Author):

The authors have addressed my criticisms and comments, and I believe that the manuscript is now ready for publication.